# Murine leukemia virus glycoGag antagonizes SERINC5 via ER-phagy receptor RETREG1

**Iqbal Ahmad[1], Jing Zhang[1], Rongrong Li[1], Wenqiang Su[1], Weiqi Liu[1], You Wu[1], Ilyas Khan[1], Xiaomeng Liu[1], Lian-Feng Li[1], Sunan Li[1]\*, Yong-Hui Zheng[2]\***

**1** State Key Laboratory for Animal Disease Control and Prevention, Harbin Veterinary Research Institute, Chinese Academy of Agricultural Sciences, Harbin, China, **2** Department of Microbiology and Immunology, University of Illinois, Chicago, Illinois, United States of America

\* lisunan@caas.cn (SL); zhengyh@uic.edu (Y-HZ)

## Abstract

Serine incorporator 5 (SERINC5) is a host restriction factor that inhibits the infectivity of certain enveloped viruses, including human immunodeficiency virus type 1 (HIV-1) and murine leukemia virus (MLV), by incorporating into the viral envelope and blocking viral entry. To counteract this, HIV-1 and MLV encode accessory proteins—Nef and glycoGag, respectively—that downregulate SERINC5 expression in producer cells. Here, we demonstrate that glycoGag employs more complex and effective mechanisms than Nef to antagonize SERINC5. Despite being a type II integral membrane protein, glycoGag primarily localizes to the cytoplasm, while Nef is mainly associated with the plasma membrane. Additionally, glycoGag is rapidly degraded by proteasomes, in contrast to the greater stability of Nef, and becomes stabilized after binding to SERINC5. While both proteins downregulate SERINC5 at the cell surface, glycoGag also targets SERINC5 at the endoplasmic reticulum (ER). We further show that this ER-specific downregulation is mediated by reticulophagy regulator 1 (RETREG1), an ER-phagy receptor, through micro-ER-phagy. These findings reveal that retroviruses hijack a selective autophagy pathway to counteract host restriction and promote productive infection.

## Author summary

HIV-1 Nef and MLV glycoGag are unrelated viral proteins, yet both counteract the same host restriction factor, SERINC5, to facilitate productive infection. In this study, we report a novel pathway through which glycoGag downregulates SERINC5. We demonstrate that while Nef downregulates SERINC5 only after it has trafficked to the cell surface, glycoGag can directly downregulate SERINC5 in the cytoplasm before it reaches the plasma membrane. Furthermore, we show that this pathway is mediated by the ER-phagy receptor RETREG1, which targets SERINC5 for degradation via micro-ER-phagy. These findings reveal that

**Data availability statement:** The authors confirm that all data underlying the findings are fully available without restriction. All relevant data are included in the paper and its Supporting information files. The raw mass spectrometry data have been deposited in iProx as project ID IPX0010260000 (https://www.iprox.cn).

**Funding:** S.L. is supported by grants from the National Natural Science Foundation of China (32172836, 32473116), Natural Science Foundation of Heilongjiang Province (YQ2024C034). J.Z is supported by a grant from the National Natural Science Foundation of China (32300129). Y.H.Z. received a salary from the National Institutes of Health (AI145504). The funders had no role in study design, data collection and analysis, decision to publish, or preparation of the manuscript.

**Competing interests:** The authors have declared that no competing interests exist.

retroviruses have evolved complex mechanisms to counteract SERINC5, highlighting the critical role of SERINC5 in restricting retroviral infections.

## Introduction

Retroviruses, in particular, lentiviruses, express accessory proteins to antagonize host restrictions and establish productive infection. Negative factor (Nef) and glycosylated Gag (glycoGag) are two unrelated accessory proteins expressed from human immunodeficiency virus type 1 (HIV-1) or murine leukemia virus (MLV), but antagonize the same host restriction factor, serine incorporator 5 (SERINC5), that strongly blocks virus entry [1].

SERINC5 (Ser5) was originally discovered in human T cells that restricts HIV-1 infection in a Nef-sensitive manner [2,3]. Ser5 also restricts MLV and other enveloped viruses [4–11]. Human Ser5 is a $\sim$45-kDa integral membrane protein with ten transmembrane helices organized into two subdomains [12]. The *trans*-Golgi network (TGN)-located cullin-3 (Cul3)-Kelch-like protein 20 (KLHL20) E3 ubiquitin (Ub) ligase polyubiquitinates Ser5 on Lysine 130 (K130) via K33/K48-branched Ub chains [13]. The K33-linked Ub chains direct Ser5 from TGN to the plasma membrane [13], where Ser5 is incorporated into the budding virus particles to inhibit the virus entry [14–17].

Nef internalizes Ser5 from the cell surface via receptor-mediated endocytosis and targets Ser5 to the Rab5$^+$ early, Rab7$^+$ late, and Rab11$^+$ recycling endosomes, resulting in degradation in endolysosomes [18]. Nef downregulation of Ser5 is dependent on the Ser5 intracellular loop 4 (ICL4) [19]. Nef interacts with the Cyclin K (CCNK)/Cyclin-dependent kinase 13 (CDK13) complex that phosphorylates Ser5 on Serine 360 (S360) in ICL4 [20]. Phosphorylated Ser5 is endocytosed and targeted to endolysosomes for degradation via the K48-linked polyubiquitination. GlycoGag could also downregulate Ser5, but the precise mechanism has not been explored as completely as Nef [5,9].

ER-phagy, often used interchangeably with reticulophagy, is a selective autophagy of the endoplasmic reticulum (ER) that manages ER quality control and size by degrading and recycling portions of the ER. Autophagy is divided into macroautophagy/autophagy, microautophagy, and chaperone-mediated autophagy (CMA) in mammalian cells [21]. Macroautophagy sequesters cargos into autophagosomes, which undergo a series of fusion processes with the late endosomes/lysosomes and mature into functional autolysosomes for bulk degradation. This process is mediated by the autophagosome biosynthesis machinery and the MAP1LC3/LC3 lipidation machinery that includes many autophagy-related (ATG) proteins. Microautophagy directly engulfs and sequesters protein substrates into lysosomes for degradation. CMA selects proteins with a KFERQ-related motif, which binds to the cytosolic chaperone, heat shock protein family A (Hsp70) member 8 (HSPA8), for recruitment to the lysosome surface. The HSPA8-associated cargo then binds to lysosome-associated membrane protein (LAMP) 2a, which is the CMA receptor for lysosomal entry.

Similarly, ER-phagy is also divided into three primary types, including macro-ER-phagy, micro-ER-phagy, and LC3-dependent vesicular transport. ER-phagy is initiated by ER-phagy receptors on the ER membrane, including RETREG1/FAM134B, RETREG1-2 (N-terminally truncated RETREG1), RETREG2/FAM134A, RETREG3/FAM134C, RTN3L, ATL3, SEC62, CCPG1, and TEX264 [22]. These receptors have at least one reticulon-homology domain (RHD) and/or LC3-interacting region (LIR), marking the ER portions for autophagic degradation [23].

We now report that while both Nef and glycoGag downregulate Ser5 at the cell surface via endolysosomes, glycoGag can also directly downregulate Ser5 via ER-phagy. We further identify RETREG1 as the ER-phagy receptor that targets Ser5 to micro-ER-phagy for degradation.

## Results

### MLV glycoGag antagonizes Ser5 much more efficiently than HIV-1 Nef

MLV glycoGag is an N-terminally extended form of the Gag polyprotein and is modified by three *N*-glycosylation sites (Fig 1A). Although glycoGag was reported as a type II integral membrane protein [9,24], it was also shown to adopt a type I membrane topology [25]. To clarify this discrepancy, we utilized the transmembrane hidden Markov model (TMHMM) to analyze glycoGag. The prediction indicates that glycoGag features an N-terminal intracellular domain, a C-terminal extracellular domain, and a transmembrane (TM) domain located at the junction of the leader and MA regions (Fig 1A). Because glycoGag downregulates Ser5 through its leader domain [9,24], this prediction further supports that glycoGag is a type II transmembrane protein.

We compared the Ser5 downregulation activity by Nef and glycoGag. Since glycoGag is functionally exchangeable with glycoMA [2,3], we used glycoMA in this measurement. Nef-deficient (ΔNef) HIV-1 viruses were produced from HEK293T cells after transfection with an HIV-1 proviral vector, an HIV-1 Env expression vector, a Ser5 expression vector, and a vector expressing either Nef or glycoMA. Subsequently, viral infectivity was assessed after infection of the HIV-1 luciferase-reporter cell line TZM-bl. As anticipated, Ser5 substantially reduced ΔNef HIV-1 infectivity by approximately 60-fold, an effect that was antagonized by both Nef and glycoMA (Fig 1B). However, glycoMA restored viral infectivity to a greater extent than Nef. We then expressed Ser5 with Nef and glycoMA and compared their protein expression by Western blotting (WB). We found that, despite glycoMA being expressed at approximately sevenfold lower levels than Nef, it reduced Ser5 expression at least twofold more effectively (Fig 1C). Thus, glycoMA reduces Ser5 expression much more effectively than Nef.

### MLV glycoGag is rapidly turned over in proteasomes

We assessed the stability of glycoGag protein after infection of NIH3T3 cells with authentic MLV. After 24 hours (hrs) of infection, cells were treated with cycloheximide (CHX) for 0, 0.5, 1.0, and 2.0 hrs, and viral proteins were detected by WB. Notably, glycoGag exhibited significantly less stability than Gag proteins (p65, p30), with a half-life of less than 30 min (Fig 2A, lanes 1–8). We then directly expressed Nef, glycoMA, and glycoGag in HEK293T cells and compared their stability. Both glycoMA and glycoGag had a similarly short half-life and were completely degraded within 2 hrs, while Nef had a half-life exceeding 10 hrs (Fig 2B).

To investigate the mechanism behind the rapid turnover of glycoGag, we treated cells with the proteasomal inhibitor MG132, as well as autophagy and lysosomal inhibitors, 3-methyladenine (3-MA), LY294002 (LY), and bafilomycin A1 (Baf-A1). None of these inhibitors affected Nef expression (Fig 2C, lanes 1–6). While 3-MA, LY, and Baf-A1 did not yield significant changes, MG132 markedly restored the expression of glycoMA and glycoGag (lanes 12, 18). We previously reported that Ser5 is polyubiquitinated by Cul3-KLHL20 E3 ubiquitin ligase [13]. When we silenced the expression of this E3 ligase complex using CRISPR ribonucleoprotein (RNP) complexes containing *Cul3*- or *KLHL20*-specific guide RNAs (sgRNAs), we found that while Nef expression remained unchanged, the expression levels of both glycoMA and glycoGag increased (Fig 2D). Thus, our results indicate that glycoGag is targeted to proteasomes for degradation.

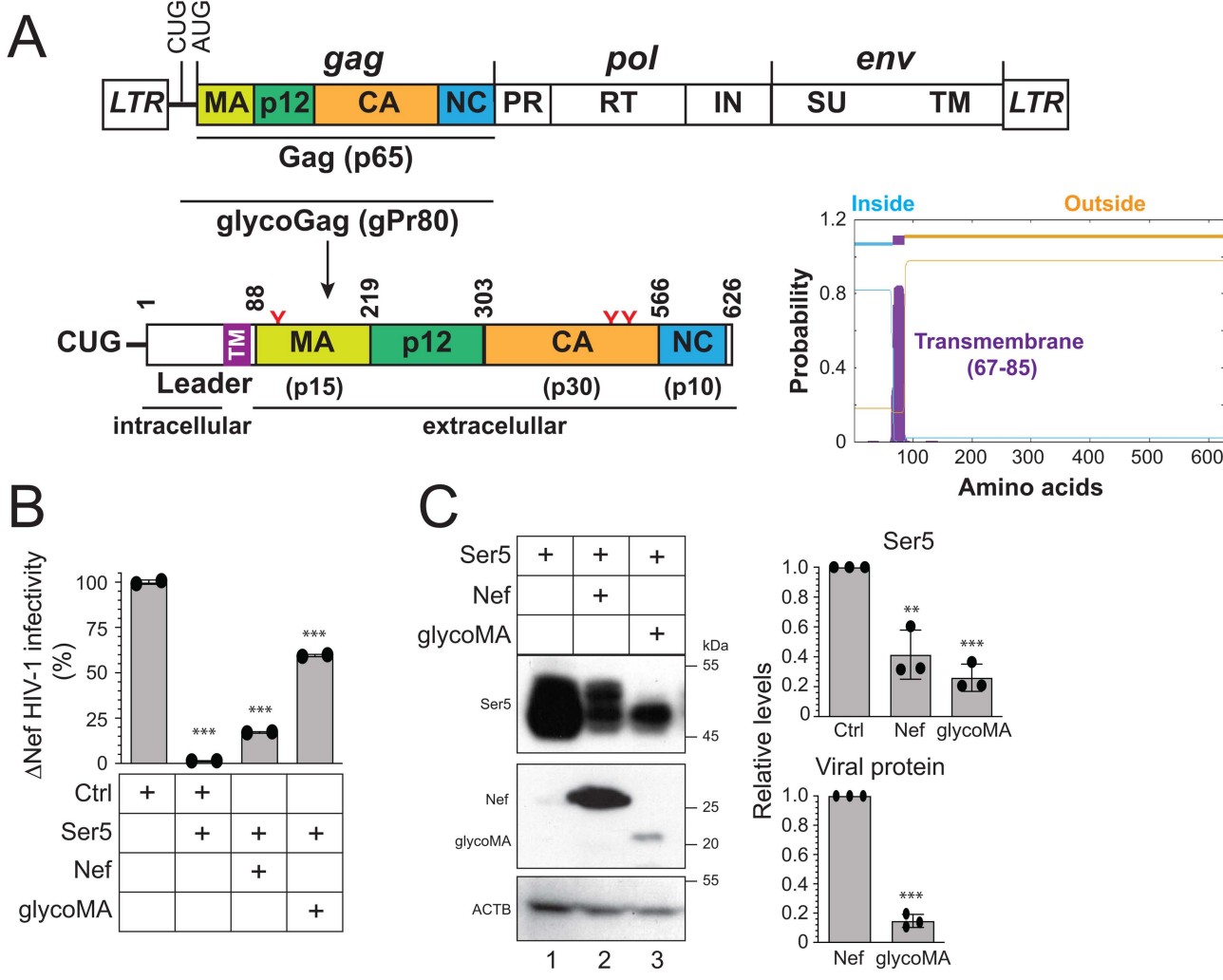

**Fig 1. MLV-glycoGag antagonizes Ser5 much more effectively than HIV-1 Nef. A)** The MLV-Gag precursor (p65) is translated from a full-length genomic RNA and cleaved into the matrix (MA), p12, capsid (CA), and nucleocapsid (NC). The glycoGag (p80) translation is initiated from a CUG codon upstream of Gag, containing three *N*-glycosylation sites (N113, N480, N505, highlighted in red). This glycoprotein is predicted to function as a type II transmembrane protein, as suggested by the transmembrane hidden Markov model (TMHMM). **B)** Nef-defective HIV-1 were produced from HEK293T cells after transfection with 1 μg Nef and Env-deficient HIV-1 proviral pNL-ΔEΔN vector, 500 ng HIV-1 Env expression vector pNLnΔBS, 50 ng pCMV6-Ser5, and 3 μg pcDNA3.1-Nef or pcDNA3.1-glycoMA. After being normalized by p24$^{Gag}$ ELISA, viral infectivity was measured via infection of TZM-bl cells. Results were shown as relative values, with the infectivity of viruses produced alone set as 100. **C)** HEK293T cells were transfected with a Ser5, Nef, and glycoMA expression vector, and protein expression was determined by western blotting (WB) using an anti-HA antibody. Levels of Ser5, Nef, and glycoMA protein expression were quantified by ImageJ and are shown as relative values, with the value of Ser5 alone or Nef set as 1. Error bars represent standard error of measurements (SEMs) calculated from two or three experiments. $n=2$ **(B)** or $n=3$ **(C)**; One-way ANOVA test: ns, not significant; *$p<0.05$, **$p<0.01$, ***$p<0.001$. Unless indicated, all experiments were repeated three times, and representative experiments are shown.

## Ser5 stabilizes glycoGag

We began by comparing the subcellular localization of Nef and glycoGag using confocal microscopy. While Nef was predominantly associated with the plasma membrane, both glycoMA and glycoGag were localized in the cytoplasm and enriched in punctate areas (Fig 3A). To confirm this observation, we isolated cytoplasmic (Cyto) and membrane (Mem) fractions from cells expressing Ser5, Nef, and glycoMA, detecting their expression in these fractions via WB. Actin was not

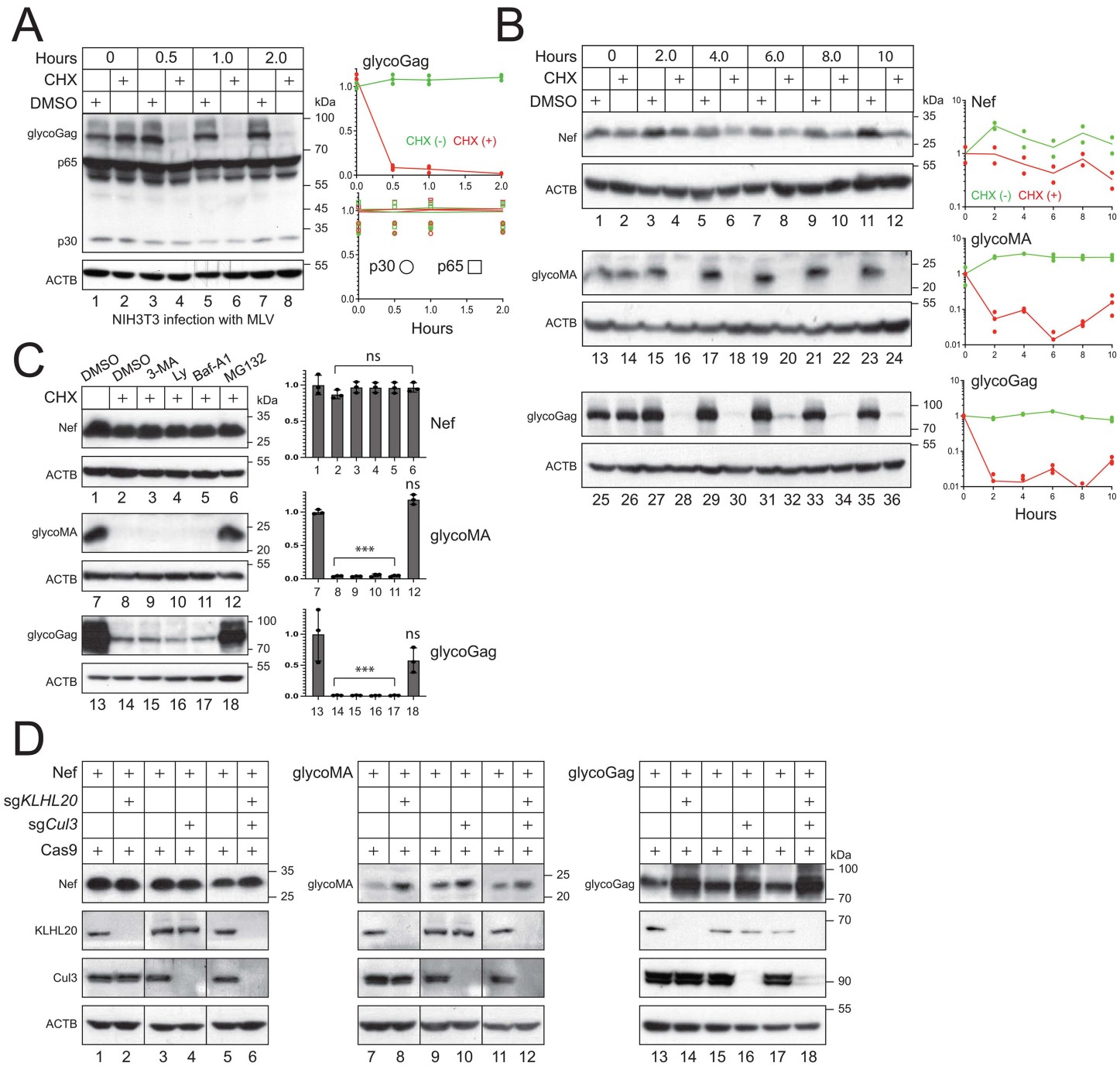

**Fig 2. MLV glycoGag is rapidly turned over in proteasomes. A)** NIH3T3 cells were infected with MLV. After 24 hrs of infection, cells were treated with 50 μM CHX or dimethyl sulfoxide (DMSO) as a vehicle control. Cells were collected at the indicated time points and the protein expression was analyzed by WB. Levels of viral proteins (glycoGag, p30, p65) were quantified using ImageJ. Results are shown as relative values, with the values at time zero set as 1. **B)** Nef, glycoMA, and glycoGag were expressed in HEK293T cells and treated with 50 μM CHX or DMSO. Their expression and stability were determined and quantified similarly. **C)** Nef, glycoMA, and glycoGag were expressed in HEK293T cells and treated with 10 mM 3-methyladenine (3-MA), 20 μM LY294002 (Ly), 100 nM bafilomycin A1 (Baf-A1), or 20 μM MG132. Their expression was analyzed by WB and quantified by Image J. Results are shown in relative values, with the values in the absence of CHX treatment set as 1. **D)** Nef, glycoMA, and glycoGag were expressed with Cas9 in HEK293T cells in the presence of *Cul3*- and/or *KLHL20*-specific sgRNAs. Protein expression was determined by WB. Error bars represent SEMs calculated from three experiments. $n = 3$ **(C)**; One-way ANOVA test: ns, not significant; *$p < 0.05$, **$p < 0.01$, ***$p < 0.001$. Unless indicated, all experiments were repeated three times, and representative experiments are shown.

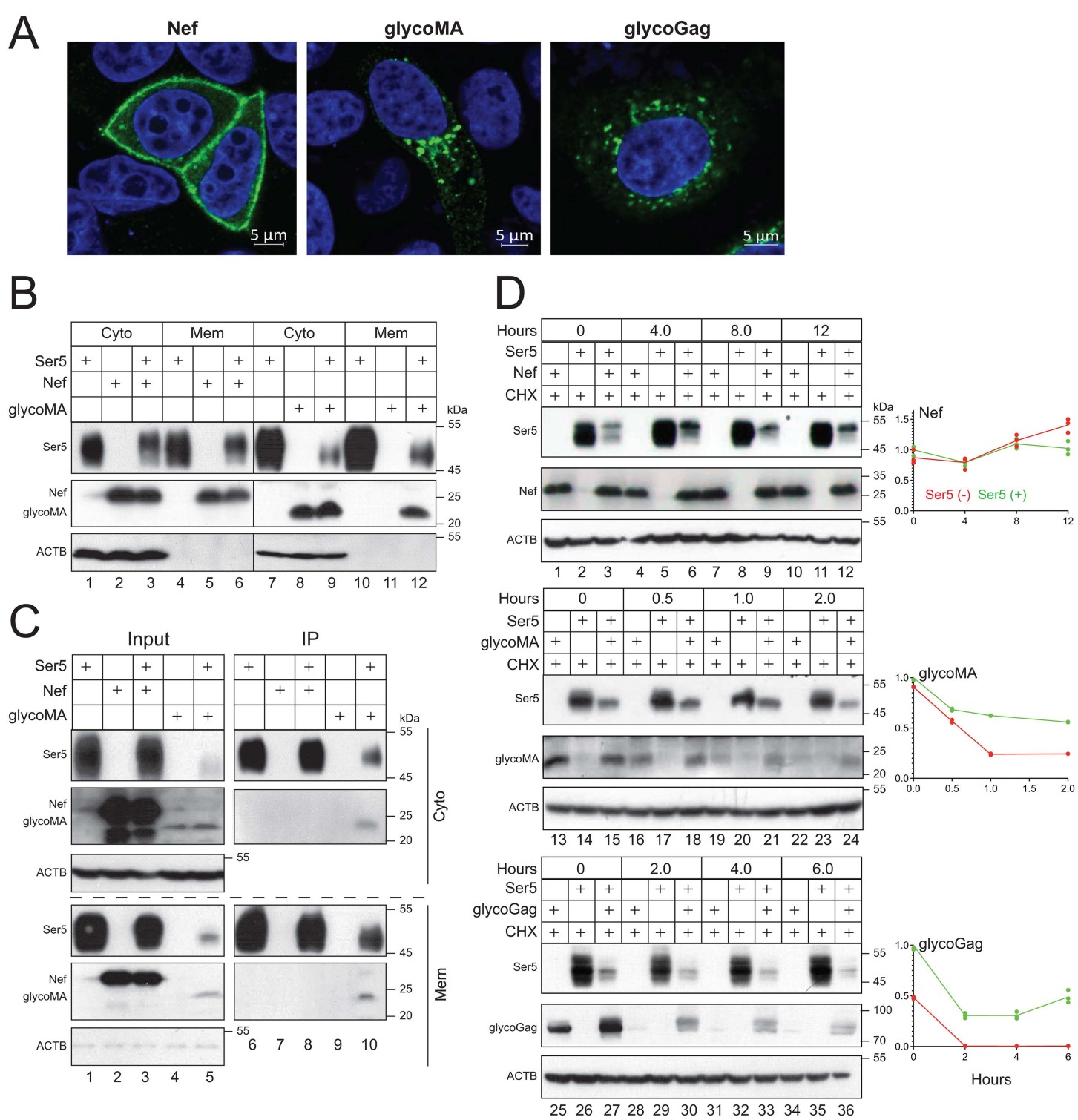

**Fig 3. Ser5 stabilizes glycoGag by targeting it to the plasma membrane. A)** Nef with a C-terminal GFP-tag, glycoMA with a C-terminal HA-tag, and glycoGag with a Myc-tag in the p12-coding region were expressed in HeLa cells. After 24 hrs, cells were stained with DAPI for the nuclei and with anti-HA or anti-Myc followed by Alexa Fluor 488-conjugated goat anti-mouse antibodies. Nef, glycoMA, and glycoGag expression were detected by confocal microscopy (scale bar 5 µm). **B)** Ser5-FLAG was expressed with Nef-HA or glycoMA-HA in HEK293T cells. Cells were separated into cytoplasmic

(Cyto) and plasma membrane (Mem) fractions, where Ser5, Nef, and glycoMA expression were detected by WB. **C)** Experiments were repeated as in **(B)**. Proteins were immunoprecipitated (IP) from Mem and Cyto fractions by anti-FLAG beads and detected by WB. **D)** Nef, glycoMA, and glycoGag were expressed alone or with Ser5 in HEK293T cells. After treatment with 50 µM CHX, cells were collected at the indicated time points and protein expression was analyzed by WB. Levels of Nef, glycoMA, and glycoGag expression were quantified using ImageJ. Results are shown as relative values, with the values at time zero in the presence of Ser5 set as 1. Unless indicated, all experiments were repeated three times, and representative experiments are shown.

detected in the Mem fraction, confirming the purity of these fractions. Ser5 and Nef were found in both fractions (Fig 3B, lanes 1, 2, 4, 5), whereas glycoMA was only detected in the Cyto fraction (lanes 8, 11). We then determine whether Ser5 affects the subcellular distribution of glycoMA. When co-expressed with Nef or glycoMA, Ser5 did not alter the levels of Nef in the cytosolic and membrane fractions (lanes 5, 6); however, it significantly increased the levels of glycoMA in the membrane fraction (lanes 11, 12).

Previously, we identified a strong interaction between Ser5 and glycoMA, but not with Nef, through co-immunoprecipitation (co-IP) [5]. We further conducted co-IP experiments to examine the interactions of Ser5 with Nef and glycoMA. While Ser5 could not pull-down Nef in the Cyto or Mem fraction, it effectively pulled glycoMA from both fractions (Fig 3C).

We then tested whether Ser5 affects the stability of glycoGag, glycoMA, and Nef after they were co-expressed in HEK293T cells. Ser5 did not alter Nef stability (Fig 3D, lanes 1–12); however, it significantly stabilized glycoGag and glycoMA, increasing their half-lives to over 2 hrs (lanes 13–24, 25–36).

## MLV glycoGag downregulates Ser5 in the cytoplasm

Ser5 K130 is located at the border between the third transmembrane domain (TMD3) and the first intracellular loop (ICL1). We reported that K130 is polyubiquitinated by the Cul3-KLHL20 E3 ubiquitin ligase, which is required for Ser5 trafficking to the plasma membrane [13]. Unlike the wild-type (WT) Ser5, its K130R mutant is dissociated from the plasma membrane and distributed in the cytoplasm. K130 is also conserved in murine Ser5 (mSer5) as K131 (Fig 4A). When mSer5 K131R mutant was created and compared with Ser5 K130R, both mutants exhibited a cytoplasmic distribution, whereas their WT counterparts were associated with the plasma membrane, as shown by confocal microscopy (Fig 4B). Following cell fractionation into Cyto and Mem fractions, both K130R and K131R were found in the Cyto but not Mem fraction (Fig 4C). In addition, they both lost the antiviral activity against glycoGag-deficient authentic MLV (MLVΔglycoGag) and Nef-deficient HIV-1 (HIV-1ΔNef) (Fig 4D). Thus, K131 is functionally conserved in mSer5.

Next, we tested how glycoGag downregulates these lysine mutants. Ser5 and mSer5 contain a total of 19 or 23 lysine residues, respectively. All these lysine residues were mutated to arginine, generating two other lysine mutants, Ser5 19K/R and mSer5 23K/R. When FLAG-tagged Ser5 (WT, K130R, 19K/R) and mSer5 (WT, K131R, 23K/R) proteins were co-expressed with HA-tagged HIV-1 Nef or MLV glycoMA, Nef downregulated only the WT proteins but not any lysine mutants (Fig 4E, lanes 2, 5, 8, 11, 14, 17). In contrast, glycoMA downregulated all these Ser5 proteins, including all these lysine mutants (lanes 3, 6, 9, 12, 15, 18). Additionally, we found that glycoMA still effectively downregulated Ser5 and K130R when Cul3 and KLHL20 were knocked down by their CRISPR RNP complexes, whereas Nef downregulation of Ser5 was effectively blocked (Fig 4F), further confirming these results.

We detected glycoMA and Ser5 interaction via IP (Fig 3C). Next, we tested whether it still interacts with the K130R mutant and whether glycoGag interacts with mSer5 and its K131R mutant through IP. We found that while Nef did not, glycoMA pulled down K130R (Fig 4G, lanes 5, 6). Additionally, glycoGag also pulled down mSer5 and K131R (lanes 12, 14). Collectively, these results demonstrate that while Nef strictly downregulates Ser5 at the cell surface, glycoGag can downregulate Ser5 within the cytoplasm as well.

Previously, we reported that Nef recruits the CCNK/CDK13 kinase complex to phosphorylate Ser5 on S360, which is required for Nef downregulation of the cell surface Ser5 [20]. Thus, we determined whether this S360 phosphorylation is required for glycoGag downregulation of Ser5. We expressed Ser5 and its two serine-to-alanine mutants (S360A, S249A)

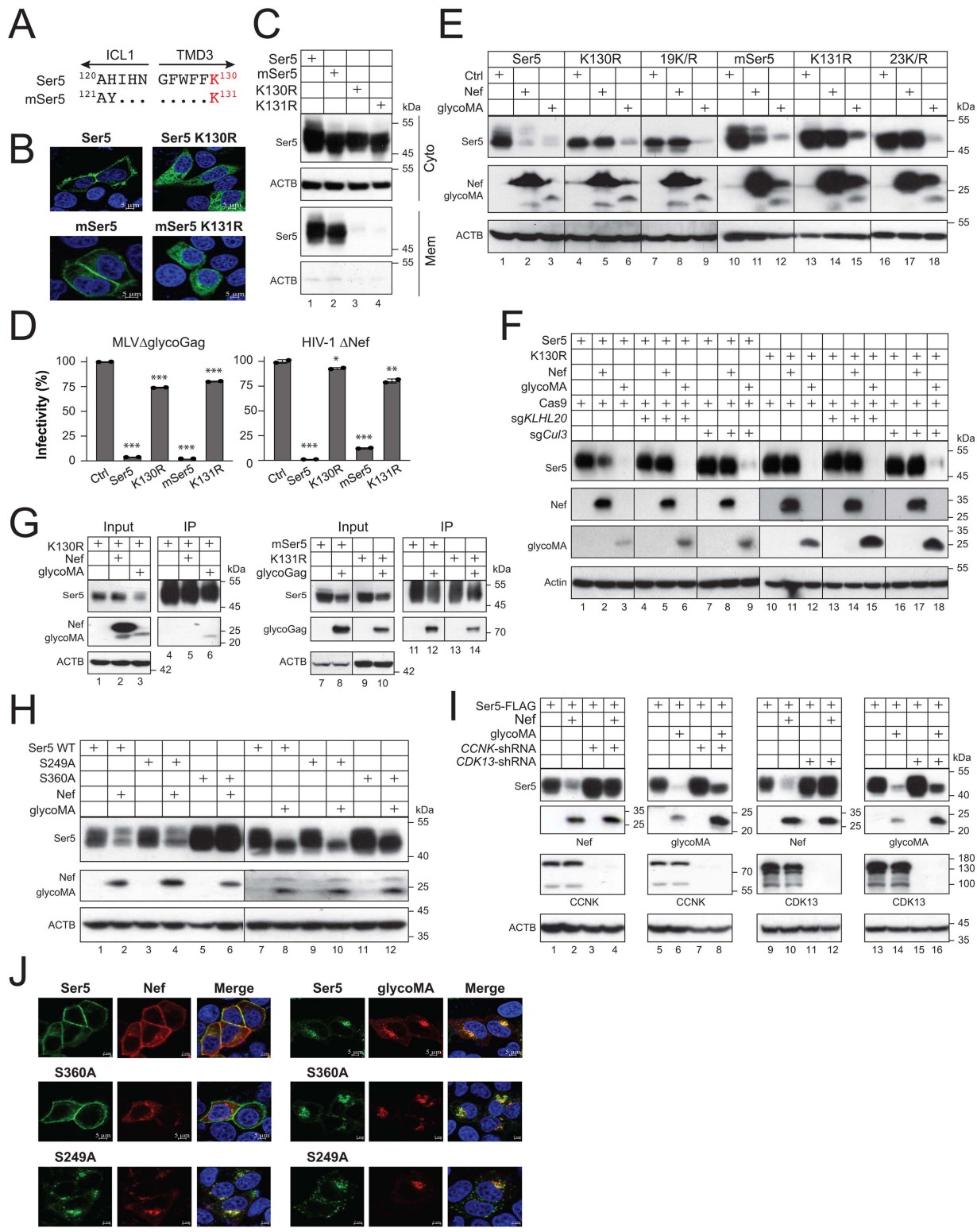

**Fig 4. MLV glycoGag downregulates Ser5 in the cytoplasm. A)** Ser5 and mSer5 amino acid sequences containing the conserved lysine (K130, K131) are aligned. The ICL1 and TMD3 regions are indicated. Dots indicate identical residues. **B)** Ser5 and mSer5 WT proteins and their lysine mutants (K130R, K131R) with a GFP-tag were expressed in HeLa cells and detected by confocal microscopy (scale bar 5 μm). **C)** Ser5 and mSer5 proteins and

their lysine mutants were expressed in HEK293T cells and their expression in Mem and Cyto fractions were analyzed by WB. **D)** MLVΔglycoGag and HIV-1ΔNef luciferase-reporter viruses were produced from HEK293T cells in the presence of indicated Ser5 proteins. After infection of NIH3T3 or TZM-bl cells, their infectivity was compared by measuring intracellular firefly luciferase activity. Results are normalized by the p30$^{Gag}$ or p24$^{Gag}$ protein levels of virions and presented as relative values, with the infectivity of viruses produced in a control (Ctrl) vector set as 100. Error bars indicate SEMs calculated from two independent experiments. **E)** Ser5, mSer5, and their lysine mutants with a FLAG-tag were expressed with Nef or glycoMA with an HA-tag in HEK293T cells. Protein expression was analyzed by WB. 19K/R, Ser5 with all 19 lysine residues replaced with arginine; 23K/R, mSer5 with all 23 lysine residues replaced with arginine. **F)** Ser5 and K130R were expressed with Nef or glycoMA in HEK293T cells, where Cas9 and *KLHL20-* or *Cul3*-specific sgRNA was also expressed. Protein expression was detected by WB. **G)** Ser5 K130R-FLAG was expressed with Nef-HA or glycoMA-HA, and alternatively, mSer5 and its K131R mutant with a FLAG-tag were expressed with glycoGag-HA in HEK293T cells. Proteins were pulled down by anti-FLAG and analyzed by WB. **H)** Ser5 and its S249A and S360A mutants were expressed with Nef or glycoMA in HEK293T cells. Protein expression was analyzed by WB. **I)** Ser5 was expressed with a *CCNK*-shRNA or *CDK13*-shRNA expression vector and a Nef or glycoMA expression vector in HEK293T cells. Protein expression was analyzed by WB. **J)** Ser5-GFP and its mutants S360A andS249A were expressed with HA-tagged Nef or glycoMA in HeLa cells. Cells were stained with DAPI (4′,6-diamidino-2-phenylindole) for the nuclei (blue) and with anti-HA followed by Alexa Fluor 594-conjugated goat anti-mouse antibodies for viral proteins and observed by confocal microscopy (scale bar 5 μm). Error bars represent SEMs calculated from two independent experiments. $n = 2$ **(C)**; One-way ANOVA test: ns, not significant; *$p < 0.05$, **$p < 0.01$, ***$p < 0.001$. Unless indicated, all experiments were repeated three times, and representative experiments are shown.

with Nef or glycoMA in HEK293T cells, and compared the Ser5 downregulation by WB. Nef downregulated Ser5 and S249A, but not S360A, whereas glycoMA downregulated all these Ser5 proteins (Fig 4H). We then expressed Ser5 with these viral proteins in HEK293T cells, where *CCNK* or *CDK13* was silenced by short-hairpin RNAs (shRNAs). When their expression was determined by WB, Nef could no longer downregulate Ser5 in the absence of CCNK or CDK13 (Fig 4I, lanes 1–4, 9–12). Nonetheless, glycoMA still downregulated Ser5 in these cells (lanes 5–8, 13–16).

Lastly, we expressed GFP-tagged Ser5, S249A, and S360A with Nef or glycoMA in HeLa cells, and their expression was observed by confocal microscopy. As we reported already [20], Nef internalized Ser5 and S249A, but not S360A (Fig 4J, left panels). Nevertheless, glycoMA internalized all these Ser5 proteins (Fig 4J, right panels). Collectively, these results further confirm that Ser5 downregulation by glycoGag does not require Ser5 on the cell surface.

## MLV glycoGag targets cytoplasmic Ser5 to lysosomes for degradation

To understand how the cytoplasmic Ser5 is downregulated, we co-expressed Ser5 and K130R with Nef or glycoMA in HEK293T cells and treated cells with MG132 and Baf-A1. Baf-A1 completely blocked Ser5 downregulation by Nef or glycoMA, as well as K130R by glycoMA, while MG132 showed no effect (Fig 5A), indicating that both Nef and glycoMA target these Ser5 proteins to lysosomes for degradation. We next expressed GFP-tagged Ser5 or K130R with mCherry-tagged LAMP1 in the presence of Nef or glycoMA and detected their co-localization by confocal microscopy. Ser5-LAMP1 colocalization was detected by the presence of Nef and glycoMA, and K130R-LAMP1 colocalization was detected only by the presence of glycoMA, which had a Pearson correlation coefficient of above 0.7 (Fig 5B). We further expressed Ser5, Nef, glycoMA, or glycoGag with the ER marker calreticulin (CALR). We found that Nef showed minimal colocalization with CALR, compared to its primary localization to the plasma membrane (Fig 5C). Similarly, Ser5 mainly localized to the plasma membrane but also showed some intracellular colocalization with CALR. Notably, glycoGag and glycoMA strongly colocalized with CALR (PCC above 0.7).

Our findings thus far indicate that Ser5 and glycoGag can be detected in the ER, with glycoGag downregulating cytoplasmic Ser5 through lysosomal degradation. We hypothesized that glycoGag downregulation of Ser5 is mediated by ER-phagy. Thus, we employed affinity-purified (AP) mass spectrometry (MS) to identify its specific ER-phagy receptor. Ser5 with a FLAG-tag was co-expressed with glycoGag and Nef, both with an HA tag, in HEK293T cells, and proteins were purified by anti-FLAG or anti-HA beads. In a total of 5 samples analyzed, ATL3, SEC62, and TEX264 were detected in all samples (Fig 5D). RETREG2 and RETREG3 were identified in samples from cells expressing Ser5 alone, Ser5 plus Nef, and Ser5 plus glycoGag. RETREG1 and CCPG1 were detected only in samples from cells expressing Ser5 plus glycoGag. Thus, we decided to investigate how these ER-phagy receptors get involved in the Ser5 degradation.

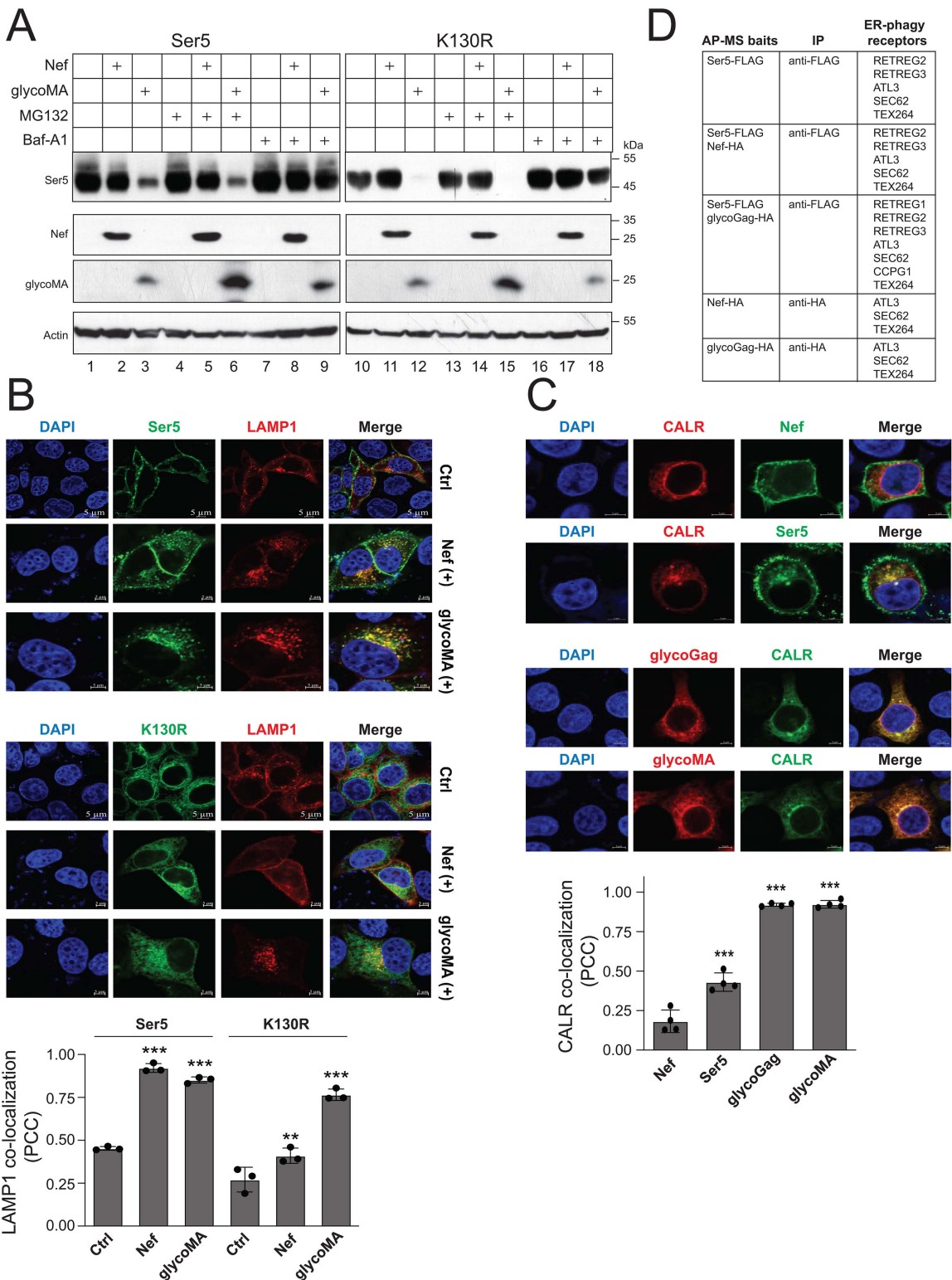

**Fig 5. MLV glycoGag targets cytoplasmic Ser5 to lysosomes for degradation. A)** Ser5 and K130R were expressed with Nef or glycoMA in HEK293T cells and treated with 20 μM MG132 or 100 nM Baf-A1. Protein expression was analyzed by WB. **B)** Ser5-GFP or K130R-GFP was expressed

with LAMP1-mCherry in HeLa cells in the presence of Nef or glycoMA. Their co-localization with LAMP1 was determined by confocal microscopy (scale bar 5 µm). Pearson's Correlation Coefficient (PCC) was calculated and shown below. C) Nef and Ser5 with a C-terminal GFP-tag were expressed with calreticulin (CALR) with a mCherry-tag in HeLa cells. Alternatively, glycoMA and glycoGag with an HA-tag were expressed with CALR with a GFP-tag in HeLa cells. GlycoMA and glycoGag were stained with mouse anti-HA followed by Alexa Fluor 594-conjugated goat anti-mouse antibodies. Co-localization of Nef, Ser5, glycoMA, and glycoGag with CALR was determined by confocal microscopy (scale bar 5 µm). PCC was calculated and shown below. D) Indicated proteins with a FLAG or HA-tag were expressed in HEK293T cells and immunoprecipitated with anti-FLAG or anti-HA beads. Affinity-purified (AP) samples were analyzed by mass spectrometry (MS). Identified ER-phagy receptors are listed. Error bars represent SEMs calculated from three experiments. $n = 3$ **(B) (C)**; One-way ANOVA test: ns, not significant; *$p < 0.05$, **$p < 0.01$, ***$p < 0.001$. Unless indicated, all experiments were repeated three times, and representative experiments are shown.

## RETREG1 enhances Ser5 downregulation by glycoGag

We reasoned that if an ER-phagy receptor is involved in Ser5 degradation, an increase in its expression should enhance this degradation. We co-expressed Ser5 with Nef and glycoGag in the presence of ectopic expression of all known ER-phagy receptors, including ATL3, RTN3L, SEC62, TEX264, CCPG1, RETREG1 (R1), RETREG1-2 (R1-2), RETREG2 (R2), and RETREG3 (R3). Our results showed that none of these ER-phagy receptors enhanced Nef downregulation of Ser5 (Fig 6A). However, RETREG1 did increase Ser5 downregulation by glycoGag, while the other receptors did not exhibit this effect (Fig 6B, lanes 23, 24; Fig 6C).

## RETREG1 is required for glycoGag downregulation of cytoplasmic Ser5

To confirm the RETREG1 activity, we chose to test the glycoGag activity in ER-phagy knockout cell lines. We have knocked out *CCPG1*, *RTN3L*, *ATL3*, and *SEC62* genes in HEK293T cells [26]. In addition, we knocked out *RETREG1*, *RETREG3*, and *TEX264* in HEK293T cells using CRISPR, which was confirmed by genomic sequencing (S1–S3 Figs). We further confirmed the *RETREG1*-knockout by WB (S4 Fig). We then expressed Ser5 K130R with glycoGag in these knockout cell lines and assessed whether glycoGag could still downregulate K130R. None of these knockouts (KOs) disrupted the glycoGag activity, except for *RETREG1*-KO (Fig 7A, lanes 3, 4). Additionally, we silenced *RETREG2* via two small interfering RNA (siRNA-*R2* #1 and #2) (S5 Fig) and found that this gene is not required for the glycoGag activity (Fig 7B). To further confirm the requirement for RETREG1, we ectopically expressed RETREG1 and its N-terminally truncated isoform (RETREG1-2), as well as RETREG2 and RETREG3, in the *RETREG1*-KO cells. Only RETREG1 restored the glycoGag activity, while the other proteins did not (Fig 7C, lanes 3, 4).

Next, we investigated how RETREG1 affects the downregulation of wild-type Ser5 by glycoGag. We co-expressed Ser5 WT and the K130R mutant with glycoGag in both HEK293T WT and *RETREG1*-KO cells. The downregulation of the K130R mutant was completely blocked in the KO cells (Fig 7D, lanes 7, 8), whereas the downregulation of Ser5 WT was only partially inhibited (lanes 3, 4). We further tested mSer5 and its K131R mutant in these cells and collected similar results (lanes 9–16). These results suggest that when the RETREG1-dependent degradation pathway is disabled, glyco-Gag can still downregulate Ser5, likely from the cell surface via endolysosomes.

Finally, we investigated the interaction between Ser5, glycoGag, and RETREG1 using co-IP. Ser5-FLAG, Nef-HA, and glycoGag-HA were expressed in HEK293T cells, and endogenous RETREG1 was pulled down using anti-FLAG or anti-HA beads. Neither Ser5, Nef, nor glycoGag alone, nor the combination of Ser5 and Nef, pulled down RETREG1 (Fig 7E, lanes 4, 5, 9, 10). However, Ser5 co-expressed with glycoGag did pull down RETREG1 (lane 6). We repeated this experiment with mSer5 and found that glycoGag could also pull down RETREG1 in the presence of mSer5 (lanes 15). These results collectively demonstrate that glycoGag targets Ser5 for ER-phagy by recruiting RETREG1.

## RETREG1 is required for MLV antagonism of Ser5

To investigate the role of RETREG1 in MLV infection, we expressed mSer5 and the K131R mutant in NIH3T3 cells, where endogenous RETREG1 was effectively knocked down using siRNAs (Fig 8A). The cells were then infected with MLV, and

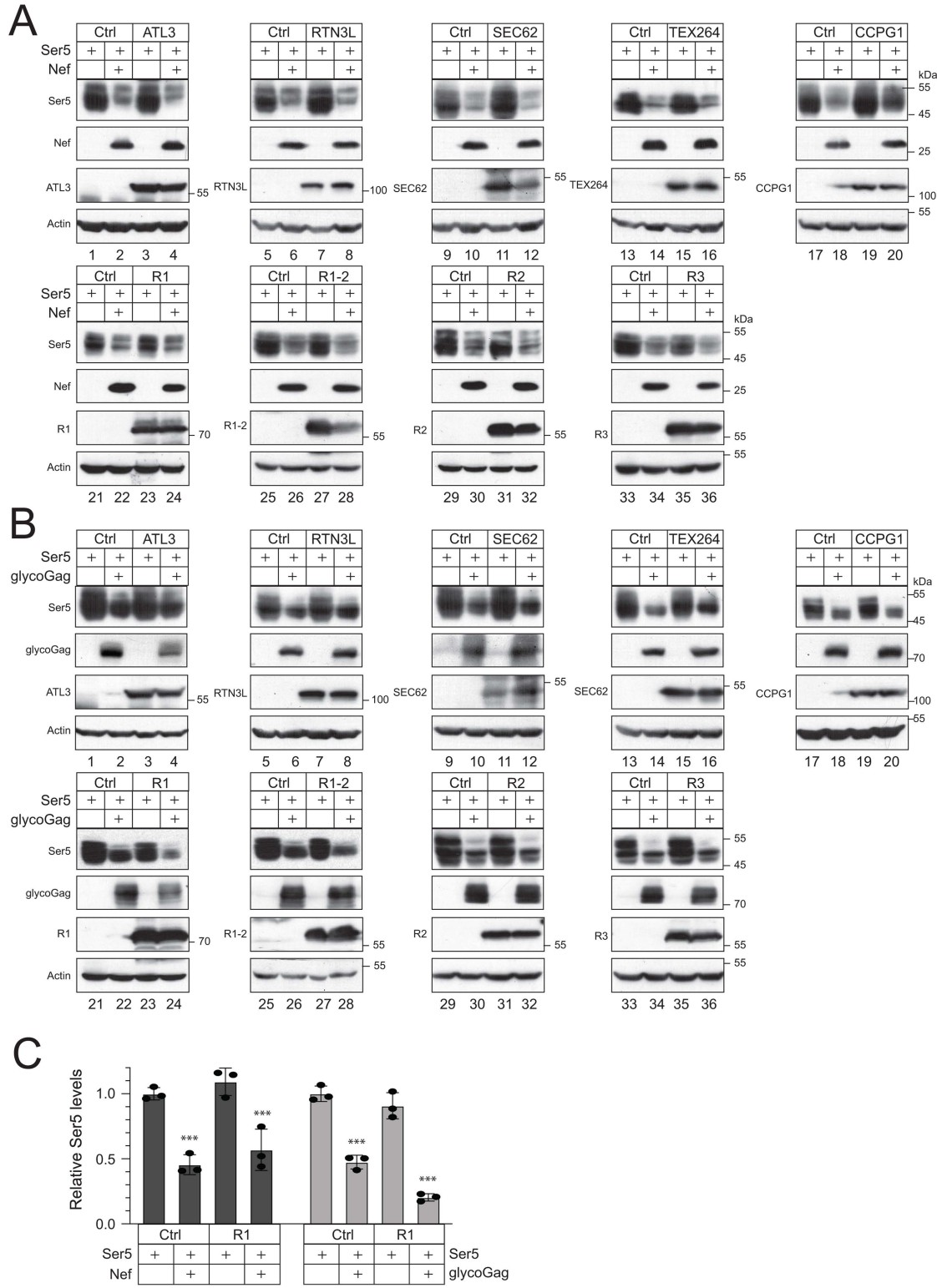

**Fig 6. RETREG1 enhances glycoGag downregulation of Ser5. A)** Ser5 was expressed with Nef in the presence of indicated ER-phagy receptors in HEK293T cells, and their expression was determined by WB. **B)** Ser5 was expressed with glycoGag in the presence of indicated ER-phagy receptors

in HEK293T cells, and their expression was determined by WB. **C)** Ser5 expression was quantified from **(A)** and **(B)** by ImageJ and presented. Error bars represent SEMs calculated from three experiments. $n = 3$ **(C)**; One-way ANOVA test: ns, not significant; *$p < 0.05$, **$p < 0.01$, ***$p < 0.001$. Unless indicated, all experiments were repeated three times, and representative experiments are shown.

Ser5 expression was analyzed by Western blot. In cells treated with control siRNAs, MLV infection efficiently downregulated both mSer5 and K131R (lanes 2, 6). In contrast, RETREG1-knockdown cells showed reduced downregulation of mSer5 (lane 4) and an almost complete loss of K131R downregulation (lane 8), consistent with prior results (Fig 7D).

Next, we produced ΔNef HIV-1 in WT and *RETREG1*-KO HEK293T cells following ectopic expression of Ser5, Nef, and/or glycoGag. Ser5 inhibited viral infectivity to a similar extent in both WT and KO cells (Fig 8B). While Nef counteracted Ser5 equally in both cell types, glycoGag was significantly less effective in *RETREG1*-KO cells than in WT cells. Together, these findings demonstrate that RETREG1 plays an important role in MLV antagonism of Ser5 during infection.

### MLV glycoGag targets Ser5 to micro-ER-phagy for degradation

To investigate the precise ER-phagy mechanism, we first determined whether it depends on autophagosomes. Thus, we tested the glycoGag activity in HEK293T cell lines where the key components of the autophagosome biosynthesis machinery, including *PIK3C3/VPS34* and *Beclin* 1 (*BECN*1), and the MAP1LC3/LC3 lipidation machinery, including *ATG3*, *ATG5*, and *ATG7*. These knockout cells were reported in our previous studies [27]. We also used another HEK293T cell line from our previous study, where the autophagy receptor *sequestosome 1* (*SQSTM1/p62*) was knocked out [28]. Ser5 WT and K130R were co-expressed with Nef or glycoMA, and their downregulation was assessed by WB. As expected, Nef did not downregulate K130R in either WT or any knockout cells (Fig 9A, lanes 7–12, 19–24, 31–36). Nef downregulated Ser5 WT in most cell lines except for *SQSTM1*- and *PIK3C3*-knockout cells (lanes 1–6, 13–18, 25–30). GlycoMA downregulated both Ser5 WT and K130R in all knockout cells (lanes 37–72). We further knocked down the CMA receptor LAMP2a using siRNAs and observed that glycoMA continued to downregulate Ser5 WT and K130R (S6 Fig). These findings suggest that Ser5 downregulation by Nef via the endolysosomal pathway requires SQSTM1/p62 and PIK3C3. Importantly, glycoGag downregulates Ser5 via micro-ER-phagy, rather than macro-ER-phagy, LC3-dependent vesicular transport, or CMA.

To further investigate this micro-ER-phagy mechanism, we tested the glycoGag activity again after treatment with three autophagy inhibitors, Baf-A1, 3-MA, and Ly. Both Nef- and glycoMA-mediated downregulation of Ser5 WT were blocked by these inhibitors (Fig 9B, lanes 1–5, 11–15). However, glycoMA-mediated downregulation of K130R was blocked by Baf-A1 and 3-MA, but not Ly (lanes 16–20). These results indicate that while PIK3C3 is not involved, an unidentified 3-MA-sensitive but Ly-insensitive factor is necessary for this micro-ER-phagy pathway.

### Discussion

We reported that glycoGag employs a mechanism similar to Nef's for downregulating Ser5 from the cell surface [9]. It was surprising to find that glycoGag does not require S360 phosphorylation for Ser5 downregulation, a modification critical for Nef's ability to downregulate Ser5. We also discovered that glycoGag effectively downregulates the Ser5 K130R mutant, indicating that it can target Ser5 in the cytoplasm. Consistently, we observed that although glycoGag is classified as a type II integral membrane protein, it primarily localizes to the cytoplasm rather than the plasma membrane. Additionally, glycoGag recruits RETREG1, an ER-phagy receptor, and targets Ser5 to the ER-phagy pathway. This suggests that glycoGag can clear Ser5 before it reaches the plasma membrane, offering another means to antagonize its antiviral activity. Thus, glycoGag employs two complementary mechanisms to degrade Ser5, a more effective antagonism than Nef (Fig 9C).

We find that, unlike Nef, glycoGag is rather short-lived, with a half-life of less than 30 min, but still antagonizes Ser5 very effectively. Unlike long-lived proteins and damaged cell organelles that are cleared by autophagy, short-lived proteins

Note: please use this file for Figure 7, and discard the previous one!!!

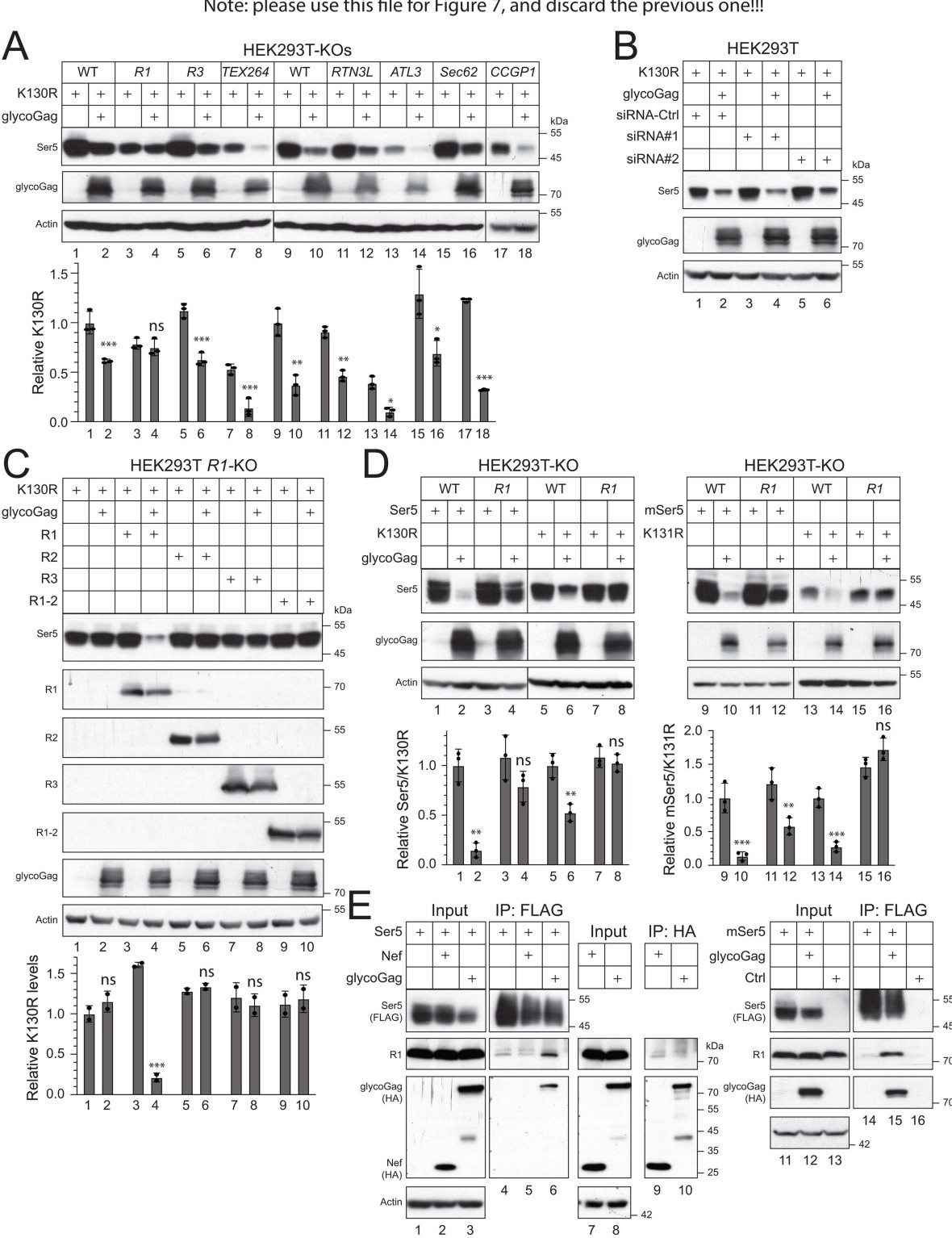

**Fig 7. RETREG1 is required for glycoGag downregulation of cytoplasmic Ser5. A)** Ser5 K130R was expressed with glycoGag in the indicated ER-phagy receptor-knockout (KO) cells, and their expression was determined by WB. **B)** K130R was expressed with glycoGag in HEK293T cells in the presence of small interfering RNA (siRNA) #1 and #2 that specifically silence RETREG2. Ser5 and glycoGag expression were determined by WB.

**C)** K130R was expressed with glycoGag in HEK293T *RETREG1* (*R1*)-KO cells in the presence of indicated ectopic ER-phagy receptors, and their expression was determined by WB. **D)** Ser5, mSer5, K130R, and K131R with a FLAG-tag were expressed with glycoGag-HA in HEK293T WT or *R1*-KO cells. Protein expression was analyzed by WB. **E)** Nef-HA and glycoGag-HA were expressed alone or with Ser5-FLAG in HEK293T cells. Alternatively, glycoGag-HA was expressed mSer5-FLAG in HEK293T cells. Proteins were pulled down by anti-FLAG or anti-HA beads. The endogenous RETREG1 in these samples was detected by WB using a specific antibody. Protein expression in **(A)**, **(C)**, and **(D)** was quantified and presented. Error bars represent SEMs calculated from two or three experiments. $n=2$ **(C)** or $n=3$ **(A)** **(D)**; One-way ANOVA test: ns, not significant; *$p<0.05$, **$p<0.01$, ***$p<0.001$. Unless indicated, all experiments were repeated three times, and representative experiments are shown.

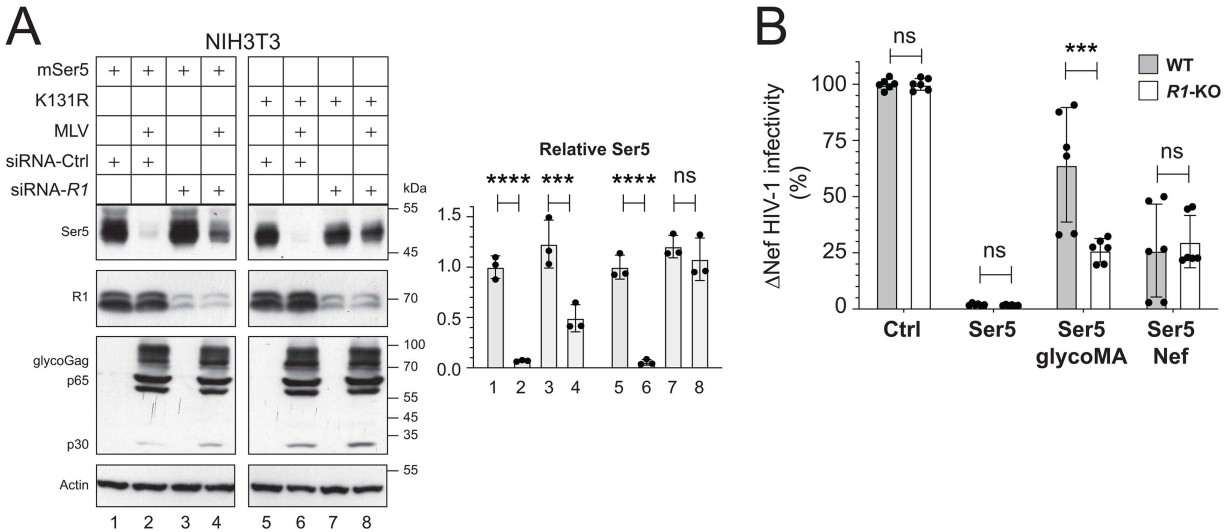

**Fig 8. RETREG1 is required for MLV antagonism of Ser5. A)** NIH3T3 cells were transfected with 40 nM siRNAs, and after 12 hrs, these cells were transfected with 200 ng Ser5 expression vector. These cells were infected with MLV, and 6 hrs later, protein expression was determined by WB 24 hrs post-infection. **B)** HEK293T WT and *RETREG1*-KO cells were transfected with 1 µg pNL-ΔEΔN vector, 500 ng pNLnΔBS, 50 ng pCMV6-Ser5, and 3 µg pcDNA3.1-Nef or pcDNA3.1-glycoGag. Viral infectivity was analyzed in TZM-bl cells and presented as before. Ser5 expression in **(A)** was quantified and presented as relative values. Error bars represent SEMs calculated from two or three experiments. $n=3$ **(A)** or $n=6$ **(B)**; One-way ANOVA test: ns, not significant; *$p<0.05$, **$p<0.01$, ***$p<0.001$. Unless indicated, all experiments were repeated three times, and representative experiments are shown.

are often degraded by proteasomes. Consistently, glycoGag is degraded in proteasomes but not lysosomes. These results are reminiscent of the HIV-1 virion infectivity factor (Vif) protein, which also has a very short half-life due to proteasomal degradation [29]. Vif antagonizes the apolipoprotein B mRNA editing enzyme, catalytic subunit 3 (APOBEC3) host restriction factors by targeting them to proteasomes for degradation [30]. It was reported that higher levels of Vif expression block the HIV-1 Gag processing, so the rapid Vif degradation of Vif could be evolutionarily beneficial to HIV-1 [31]. In addition, only a very small amount of glycoGag is needed to counteract Ser5 and another host factor IFITM3, and higher glycoGag expression causes loss of MLV infectivity [4,32]. Thus, we speculate that glycoGag may also block the MLV Gag maturation, so its rapid degradation is evolutionarily beneficial to MLV.

The endolysosomal system selects extracellular cargo via endocytosis and phagocytosis for degradation. We reported that Nef internalizes Ser5 from the cell surface via receptor-mediated endocytosis, which is subsequently recruited to lysosomes via endosomes [18]. Unlike the autophagy pathways, the endolysosmal machinery remains to be fully defined. Here, we found that this Nef activity is blocked in *SQSTM1*-KO and *PIK3C3*-KO cells, highlighting a new role of SQSTM1/p62 and PIK3C3/VPS34 in endolysosomal degradation. SQSTM1 is a macroautophagy receptor that recruits polyubiquitinated cargo to the autophagosome membrane for degradation. Thus, SQSTM1 should play a similar role in this endolysosomal pathway by recruiting polyubiquitinated Ser5 proteins during this degradation process. In addition, PIK3C3/VPS34

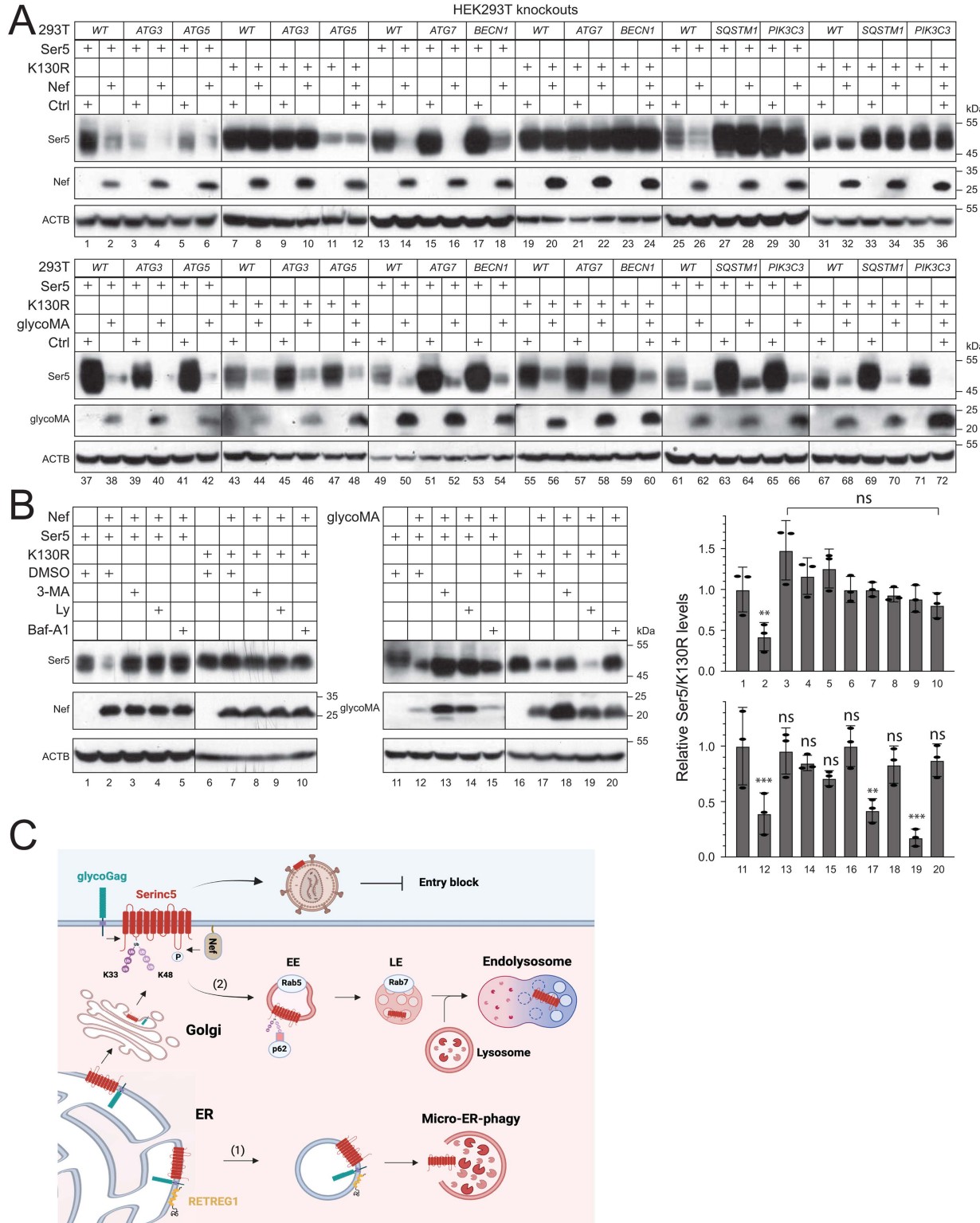

**Fig 9. MLV glycoGag targets Ser5 to micro-ER-phagy for degradation. A)** Ser5 and K130R were expressed with Nef or glycoMA in HEK293T WT and indicated knockout cells that do not express ATG3, ATG5, ATG7, BECN1, SQSTM1/p62, or PIK3C3. Protein expression was analyzed by WB. **B)** Ser5 and K130R were expressed with Nef or glycoMA in HEK293T cells and treated with 10 mM 3-MA, 20 μM Ly, or 100 nM Baf-A1. Protein expression

was analyzed by WB. **C)** A proposed model for MLV glycoGag downregulation of Ser5. GlycoGag is expressed as a type II integral membrane protein in the ER, where it interacts with Ser5. This interaction leads to the recruitment of RETREG1, resulting in Ser5 degradation via micro-ER-phagy (1). If Ser5 is not cleared within the ER, it proceeds to the Golgi apparatus and is subsequently delivered to the cell surface after undergoing polyubiquitination. Throughout this process, glycoGag is also recruited to the cell surface by Ser5. At the cell surface, both glycoGag and HIV-1 Nef can internalize Ser5 through early endosomes (EE) and late endosomes (LE), culminating in degradation within endolysosomes (2). Ser5 expression in (**B**) was quantified and presented. Error bars represent SEMs calculated from three experiments. $n = 3$ (**B**); One-way ANOVA test: ns, not significant; *$p < 0.05$, **$p < 0.01$, ***$p < 0.001$. Unless indicated, all experiments were repeated three times, and representative experiments are shown.

is responsible for producing lipid phosphatidylinositol-3-phosphate (PI3P) that is required for macroautophagy and endocytosis. PIK3C3 assembles two protein complexes with different cellular proteins, and both complexes contain BECN1. Complex I produces PI3P at the phagophore, promoting the autophagosome formation, and complex II produces PI3P at early endosomes, promoting the endocytic sorting [33]. Because BECN1 is not required for Ser5 downregulation by Nef, it is unclear how PIK3C3 gets involved in endolysosomal degradation.

RETREG1/FAM134B was the first protein identified as an ER-phagy receptor [34], which belongs to the protein family with sequence similarity 134 (FAM134), including two other conserved ER-phagy receptors FAM134A/RETREG2 and FAM134C/RETREG3 [35]. They all have an N-terminal RHD with two hydrophobic domains connected by a hydrophilic loop. Each hydrophobic domain forms a hairpin within the membrane bilayer and induces ER membrane curvature. They also have an identical LIR in the C-terminal disordered region, which binds to autophagosomes. RETREG1 is also expressed as a truncated isoform RETREG1-2/FAM134B-2, which lacks the first 141 N-terminal residues and RHD [36,37]. All these proteins are involved in starvation-induced macro-ER-phagy. Now, we show that RETREG1 is also involved in micro-ER-phagy. Because RETREG1-2 is inactive, RHD should be critical for RETREG1 to initiate this micro-ER-phagy.

Unlike macroautophagy and CMA, the mechanism for microautophagy still remains elusive. Based on the morphological changes of the lysosome, three types of microautophagy have been proposed: lysosomal protrusion, lysosomal invagination, and endosomal invagination [38]. Like CMA, the endosomal microautophagy selects proteins with a KFERQ-like motif and depends on HSPA8. However, unlike CMA, it uptakes cytoplasmic proteins in the late endosomes via the endosomal sorting complex required for transport (ESCRT) machinery. Notably, we find that the K130R degradation by glycoGag was blocked by 3-MA but not Ly. These two autophagy inhibitors inhibit class I PI3Ks, and they also have some other cellular targets [39,40]. We suggest that there is a 3-MA-sensitive but Ly-insensitive host factor that plays a critical role in microautophagy. Further characterization of this unknown factor will yield critical new insights into the molecular machineries of microautophagy.

## Materials and methods

### Cell lines

Human embryonic kidney (HEK) 293 cell line transformed with SV40 large T antigen (HEK293T), mouse embryonic fibroblasts cells (NIH3T3), and human cervical carcinoma cell line HeLa were purchased from American Type Culture Collection (ATCC) (CRL-3216, CRL-1658, CRM-CCL-2). The HIV-1 luciferase reporter TZM-bI cell line (ARP-8129) was obtained from the National Institutes of Health (NIH) HIV AIDS Reagent Program. HEK293T *ATG3*-, *ATG5*-, *ATG7*-, *BECN1*-, and *PIK3C3*-knockout cell lines were reported [27]. HEK293T *SQSTM1*-knockout cell line was reported [28]. HEK293T *RETREG1*-, *RETREG3*-, and *TEX264*-knockout cell lines were generated by CRISPR-Cas9 and confirmed by Sanger DNA sequencing, as we did previously [41,42]. The oligo sequences for single-guide RNA (sgRNAs) targeting *RETREG1*, *RETREG3* and *TEX264* are 5'-CGCGGTAACCTGGCTGCTCG-3', 5'-TTGTCTAATGCGTCGGGTCT-3', and 5'-GATAAGTGCCGATGTGCCGT-3', respectively. The generation of HEK293T *CCPG1*-, *RTN3L*-, *ATL3*-, and *SEC62*-knockout cell lines were reported [26].

All these cells were maintained in Dulbecco's modified Eagle medium (DMEM; Thermo Fisher Scientific, 11965092) supplemented with 10% fetal bovine serum (Inner Mongolia Opcel Biotechnology) and 1% penicillin-streptomycin (penstrep; Thermo Fisher Scientific, 10378016) and cultivated at 37°C in humidified atmosphere in a 5% $CO_2$ incubator.

### Antibodies and inhibitors

The mouse anti-HA (H3663), rabbit anti-FAM134B (HPA012077), and horseradish peroxidase (HRP)-conjugated anti-FLAG (A8592), anti-HA (H6533), and anti-actin (A3854) antibodies were purchased from Sigma. The mouse anti-Myc (2776S) and rabbit anti-Cullin3 (2759S) were purchased from Cell Signaling. Rabbit anti-CycK (Ab85854) was purchased from Abcam; rabbit anti-CDC2L5 (NB100-68268) and KLHL20 (NBP1-79570-20) were from Novus; rabbit anti-LAMP2a (AF1036) was from Beyotime; rabbit anti-MLV p30 gag (AP33447PU-N) was from Origene; Alexa Fluor 594-conjugated goat anti-mouse (A11032) and Alexa Fluor 488-conjugated goat anti-mouse (A11029) were from Invitrogen. 3-Methyleadenine (M9281), LY-294002 (L9908), MG132 (C2211), and cyclohexamide (C7698) were from Sigma; bafilomycin A1 (sc-201550) was from Santa Cruz Biotechnology.

### Bacterial strains

Escherichia coli (*E. coli*) HB101 cells (Promega, L2011) were used to produce HIV-1 proviral vectors. These bacteria were cultured in Luria-Bertani (LB) broth (Hopebio, HB0128) on a shaking incubator at 30°C. All other plasmids were produced from *E. coli* DH5α (Vazyme, C502) cells and were cultured in LB broth at 37°C on a shaking incubator. Penicillin 100 μg/mL or Kanamycin 50 μg/mL (Solarbio, A8180, K8020) were added to the medium for selective growth of transformed bacteria.

### Expression vectors

The Env and Nef-deficient HIV-1 proviral vector pNLΔEΔN (pNLenCAT-Xh) and HIV-1 Env expression vector pNLnΔBS were reported [43]. The MLV provirus vector (pNCA) was a gift from Stephen Goff (Addgene plasmid # 17363; http://n2t.net/addgene:17363; RRID:Addgene_17363) [44]. GlycoGag-deficient MLV proviral vector pNCAΔgG and pLNCX2-Luc were reported [9]. A codon-optimized glycoGag gene (accession no. *J02255*) was cloned into pcDNA3.1 vector via NheI/HindIII digestion, which contained a Myc- or HA-tag in the p12-coding region. pCMV-Ser5-EGFP, pCMV-Ser5-K130R-EGFP, pCMV6-Ser5-FLAG, pCMV6-Ser5-K130R-FLAG, pCMV6-Ser5-1-19K/R-FLAG, pCMV6-Ser5-S360A-FLAG, pCMV6-Ser5-S249A-FLAG, pCMV6-mSer5-FLAG, pcDNA3.1-SF2Nef-HA, and pcDNA3.1-glycoMA-HA were reported [5,9,20]. pEGFP-N1 vector expressing SF2Nef was constructed by PCR via KpnI/AgeI digestion. pCMV-LAMP1-mCherry was created by directly cloning the LAMP1 from pCMV-LAMP1-mGFP into pCMV vector with a mCherry tag. S360A and S249A mutations were also introduced into pCMV-Ser5-EGFP by site-directed mutagenesis. mSer5 K131R mutation in pCMV6-mSer5-FLAG and pCMV-mSer5-EGFP were also created by site-directed mutagenesis. Primers and cloning methods are available upon request.

pCMV6-mSer5-23K/R-FLAG, pCMV6-RETREG1-HA (Accession no. *NM_001034850.3*), pCMV6-RETREG2-HA (Accession no. *NM_001321109.2*), pCMV6-RETREG3-HA (Accession no. *NM_178126.4*), pCMV6-RETREG1-2-HA (Accession no. *NM_019000.5*), pCMV6-SEC62-HA (Accession no. *NM_003262.4*), and pCMV6-TEX264-FLAG (Accession no. *NM_015926.6*) were purchased from Gensoul Technology. pCMV6-mSer5-23K/R-FLAG, pCMV6-RTN3L-FLAG (Accession no. *NM_001265590*), pCMV6-ATL3-FLAG (Accession no. *NM_015459*), and pCMV6-CCPG1-FLAG (Accession no. *NM_004748*) were purchased from Comate Biosciences. pEGFP-N1 vector expressing CALR was constructed by PCR via XhoI/BspEI digestion. pCAGGS-CALR-mCherry was constructed by cloning CALR into the pCAGGS-mCherry vector via EcoRI digestion. pMJ920 Cas9 was a gift from Jennifer Doudna (Addgene plasmid # 42234) [45]. pGEM-T vectors expressing *KLHL20* and *Cul3* sgRNAs were reported previously [13]. pLKO.1-CCNKshRNA and pLKO.1-CDK13shRNA vectors were reported previously [20].

## Synthesis of siRNAs

Human *LAMP2a* (5'-GCACCCAUGCUGGAUAUTT-3'/5'-AUAUCCAGCAUGAUGGUGCTT-3'), human *RETREG2* (5'-CCCUGGUGGUUUAUCAUGATT-3'/5'-UCAUGAUAAACCACCAGGGTT-3'), murine *RETREG1* (5'-GCCAAAGAGUUAUCUGUGUTT-3'/5'-ACACAGAUAACUCUUUGGCTT-3'), and control (5'-UUCUCCGAACGUGUCACGUdTdT-3'/5'-ACGUGACACGUUCGGAGAAdTdT-3') siRNAs were synthesized by GenePharma.

## Viral infectivity determination

Nef-deficient HIV-1 for a single round of replication was produced by transfecting HEK293T cells with pNLΔEΔN and pNL-nΔBS in the presence of Ser5 expression vectors using polyethylenimine (PEI) as transfection reagent. Viruses were collected after 48 hrs and quantified with p24$^{Gag}$ ELISA. Viral infectivity was determined by infecting TZM-b1 cells in triplicate in 96-well plates for 48 hrs. Cells were lysed and luciferase activities were determined using the Firefly Luciferase Assay Kit (UElandy, F6024XL). Infectivity was calculated after being normalized by p24$^{Gag}$ [46].

GlycoGag-deficient MLV was produced from HEK293T cells after transfection with pNCAΔgG and pLNCX2-Luc in the presence of Ser5 expressing vectors. After 48 hrs, virions were purified by ultracentrifugation and quantified by p30$^{Gag}$ via WB. Viral infectivity was determined by intracellular luciferase activity after infection of NIH3T3 cells, which was normalized by p30$^{Gag}$.

## Protein stability analysis

MLV was produced from HEK293T cells after transfection with pNCA for 48 hrs and used to infect NIH3T3 cells cultured in 6-well plates. Alternatively, HEK293T cells were transfected with a glycoGag, glycoMA, and Nef expression vector using PEI as transfection reagents. After 24 hrs of infection or transfection, cells were treated with 50 μM CHX or DMSO for different times and analyzed by WB. To detect viral protein degradation pathway, 293T cells were transfected with Nef, glycoGag, and glycoMA expression vectors. After 24 hrs, cells were treated with 10 mM 3-methyladenine (3-MA), 20 μM LY294002 (Ly), 100 nM bafilomycin A1 (Baf-A1), or 20 μM MG132 for 4 hrs. Cells were lysed, and protein expression was analyzed by WB.

## Western blotting (WB)

HEK293T cells were seeded either in 6-well plates or 10-cm dishes, and the next day, transfected with expression vectors according to the experimental design. After 24 hrs of transfection, cells were either lysed with RIPA buffer containing protease inhibitors (Target Mol, USA) for protein extraction or collected for extraction of membrane proteins. The Membrane and Cytosol Protein Extraction Kit (Beyotime, P0033-1) was used for membrane protein extraction. Total cell lysate and membrane proteins were then applied to sodium dodecyl-sulfate polyacrylamide gel electrophoresis (SDS-PAGE) and then blotted on polyvinylidene difluoride (PVDF) membranes (Sigma, ISEQ00010). Membranes were then blocked with 5% skimmed milk powder dissolved in TBST (Tris-buffered saline), and incubated with HRP-conjugated anti-FLAG, anti-HA, and anti-actin (1:5000) antibodies. Alternatively, membranes were first incubated with primary antibodies, including anti-FAM134B (1:3000), anti-Cullin3 (1:1000), anti-CycK (1:2000), anti-CDC2L5 (1:2000), anti-Myc (1:3000), anti-KLHL20 (1:2000), anti-LAMP2a (1:3000), and anti-MLV p30 gag (1:10000). After being washed with TBST, proteins were incubated with HRP-conjugated anti-mouse or rabbit secondary antibodies (1:3000). The blotted proteins were detected by enhanced chemiluminescence substrate (Applygen, P1010) using X-ray films (FUJI) as reported [47]. Films were scanned, and protein bands were quantified by ImageJ. Adobe Photoshop and Adobe Illustrator were used to generate the figures.

## Immunoprecipitation

To detect RETREG1 interactions with Ser5, Nef, and glycoGag, FLAG-tagged Ser5 or HA-tagged Nef, or glycoGag were expressed or co-expressed in HEK293T cells. After 24 hrs of transfection, cells were lysed with RIPA buffer and proteins

were precipitated with anti-FLAG M2 magnetic beads (Sigma, M8823) or anti-HA Affinity Gel (Beyotime P2287). Cell lysate (Input) and immunoprecipitated (IP) samples were analyzed by WB [42].

## Confocal microscopy

HeLa cells with an initial density of $1.5 \sim 2.0 \times 10^5$/dish were seeded on poly-L-lysine-coated coverslips and transfected with indicated vectors using Lipofectamine 3000. After 24 hrs, cells were washed with phosphate buffer saline (PBS) and fixed with 4% paraformaldehyde for 5 min. Cells were then permeabilized with 0.1% Triton X-100 for 10 min, followed by 2 hrs' blocking with 10% FBS. Cells were then incubated with a mouse anti-HA antibody (1:1000) overnight at 4°C, washed, and coated with secondary antibodies (1:1000), including Alexa Fluor 594-conjugated goat anti-mouse or with Alexa Fluor 488-conjugated goat anti-mouse for 1 hr. Cells were washed, and the nuclei were stained with 4′,6-diamidino-2-phenylindole (DAPI) (Sigma, D9542) for 30 min, followed by washing. The fluorescence signals were then visualized by confocal microscope (ZEISS, LSM880). Image J software was used for quantitative measurement of colocalization. Colocalization correlation of the intensity distributions between two channels was determined by calculating Pearson's correlation coefficient (PCC).

## Mass spectrometry

HEK293T cells were transfected with the Ser5-FLAG, Nef-HA, and glycoGag-HA expression vectors. After 24 hrs of transfection, cells were lysed with RIPA buffer. The cell lysate was incubated with anti-FLAG M2 magnetic beads or anti-HA Affinity Gel for 24 hrs at 4°C. The beads and the gel were then washed with PBS five times and mixed with protein loading buffer (5X) and electrophoresed in SDS-PAGE. The gel was then stained with Coomassie blue and sent to the Laboratory of Proteomics, Institute of Biophysics, Chinese Academy of Sciences, for Nano LC-MS/MS and database search analysis. The raw data can be downloaded from iProX (project ID: IPX0010260000).

## Statistical analysis

All experiments were performed independently at least three times unless indicated. SPSS Statistics Software (Version 23; IBM, Inc., New York, USA) was used for the data analysis. Quantitative values of data were expressed as mean ± standard error of measurements (SEMs) and represented by error bars. Statistical comparisons were performed using one-way analysis of variance (ANOVA) followed by the LSD post-hoc test. A p-value less than 0.05 was considered statistically significant. Significance levels are indicated as follows: $*p < 0.05$, $**p < 0.01$, $***p < 0.001$; ns indicates not significant ($p > 0.05$).

## Supporting information

**S1 Fig. Validation of HEK293T *RETREG1* (*R1*)-KO cells by genomic sequencing.**
(PDF)

**S2 Fig. Validation of HEK293T *RETREG3* (*R3*)-KO cells by genomic sequencing.**
(PDF)

**S3 Fig. Validation of HEK293T TEX264-KO cells by genomic sequencing.**
(PDF)

**S4 Fig. Validation of *R1*-KO clones by Western blotting.**
(PDF)

**S5 Fig. Validation of *RETREG2* siRNAs.** RETREG2 was expressed with its specific siRNAs (#1, #2) or a control (Ctrl) in HEK293T cells and its expression was detected by WB.
(PDF)

**S6 Fig. LAMP2a is not required for glycoMA downregulation of Ser5.** Ser5 and K130R were expressed with glycoMA in HEK293T cells in the presence of LAMP2a-specific siRNA or its control (Ctrl). Protein expression was detected by WB. (PDF)

## Acknowledgments

We thank Huiling Ren, Xiaojun Wang, Stephen Goff, and Jennifer Doudna for the reagents. We thank Zhigao Bu for suggestions on this project. We thank Jifeng Wang from the Laboratory of Proteomics, Institute of Biophysics, Chinese Academy of Sciences, for mass spectrometry analysis.

## Author contributions

**Conceptualization:** Yong-Hui Zheng.

**Data curation:** Iqbal Ahmad, Sunan Li.

**Formal analysis:** Sunan Li, Yong-Hui Zheng.

**Investigation:** Iqbal Ahmad, Sunan Li, Rongrong Li, Weiqi Liu, You Wu, Ilyas Khan, Xiaomeng Liu, Lian-Feng Li.

**Project administration:** Sunan Li.

**Resources:** Jing Zhang, Wenqiang Su.

**Writing – original draft:** Yong-Hui Zheng.

**Writing – review & editing:** Yong-Hui Zheng.

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
