## [Decision Letter · Decision Letter 0]

23 Apr 2025

PPATHOGENS-D-25-00567

Murine Leukemia Virus GlycoGag Antagonizes SERINC5 via ER-phagy Receptor RETREG1

PLOS Pathogens

Dear Dr. Zheng,

Thank you for submitting your manuscript to PLOS Pathogens. As with all papers, your manuscript was reviewed by members of the editorial board. Based on our assessment, we have decided that the work does not meet our criteria for publication and will therefore be rejected. If external reviews were secured, reviewers' comments will be included at the bottom of this email.

We are sorry that we cannot be more positive on this occasion. We very much appreciate your wish to present your work in one of PLOS's Open Access publications. Thank you for your support, and we hope that you will consider PLOS Pathogens for other submissions in the future.

Yours sincerely,

Welkin E. Johnson

Academic Editor

PLOS Pathogens

Richard Koup

Section Editor

PLOS Pathogens

Michael Malim

Editor-in-Chief

PLOS Pathogens

orcid.org/0000-0002-7699-2064

Additional Editor Comments (if provided):

Reviewers' comments:

Reviewer's Responses to Questions

**Part I - Summary**

Reviewer #1: Ahmad et al. report a mechanism enabling the retroviral accessory protein glycoGag to counteract the restriction factor SERINC5. Specifically, the authors identify that glycoGag uses a strategy to antagonize SERINC5 that is distinct from how Nef antagonizes SERINC5. The authors report that glycoGag interacts with SERINC5 in the endoplasmic reticulum and triggers SERINC5 degradation through ER-phagy. This conclusion is supported by AP-MS data showing interaction between glycoGag and ER-phagy receptors such as RETREG1. The authors also show that glycoGag loses the capacity to antagonize SERINC5 in cells deficient for RETREG1. Additionally, the authors report some interesting results regarding how Nef downmodulates SERINC5. It appears to do so in a manner that requires p62 and PI3KC3, which are proteins normally associated with macroautophagy. This study is well-written and nicely organized and presented, with carefully crafted figures. However, I believe there are a number of changes that could be introduced to improve the manuscript.

Reviewer #2: In this manuscript, Ahmad and co-authors provide details on the molecular mechanism by which glycoGag downregulates SERINC5. Strong evidence shows that the mechanism is different between Nef (the Serinc5 antagonist in HIV-1) and glycoGag (the antagonist in MLV) with data supporting a role for ER-phagy as the preferred method by which glycoGag accomplishes this. The manuscript is well-written, and the data presented is of high quality. However, the authors at times make overstatements from their data, which should be accompanied by additional experiments to unequivocally support their conclusions.

Reviewer #3: In this manuscript, the authors explore the mechanism by which the MLV glycoGag protein counteracts the antiviral effect of SERINC5. The authors show that glycoGag utilizes selective autophagy to reduce the cellular levels of SERINC5. The authors show that glycoGag is highly unstable with a very short half life and localizes in internal organelles (presumably). They provide evidence that Nef and glycoGag utilize different mechanisms to downregulate cellular levels of SERINC5 in the cell. In addition, the authors show that the glycoGag effect on SERINC5 is dependent on the host protein RETREG1, an ER-phagy receptor, and is mediated by micro-ER-phagy. However, a number of important weaknesses have been identified. The fact that the vast majority of the findings have been performed in the context of co-transfections in HEK293T cells and using the human ortholog of SERINC5 significantly weaken the relevance to natural infection of MLV. In fact, processes like autophagy are often regulated by multiple viral proteins during infection, thus by transfecting only glycoGag the authors may not be seeing what happens under physiological conditions during MLV infection. Moreover, most experiments are performed at least twice or no information is provided about the number of replicates affecting the rigor of this study. Finally, the paper needs to be better edited with missing information (e.g., Fig 5D legend) and not enough information is provided in the material and methods section (e.g., transfection conditions, antibody concentrations etc).

**Part II – Major Issues: Key Experiments Required for Acceptance**

Reviewer #1: Major:

1. In Figure 1B, Nef only minimally restores infectivity of DeltaNef HIV-1 infectivity. Authors should comment on this result. Furthermore, in Figure 1C, the authors should address levels of SERINC5 in virions, not just producer cell lysates. Many papers studying Nef/glycoGag have not observed a decrease in SERINC5 levels in producer cells. I think Figure 1D should be removed, since it is an arbitrary, author-defined definition of “antagonism” that takes into account detected Nef and glycoMA protein. It is unnecessary to make a figure to make this point.

2. When addressing glycoGag stability and activity, including where it localizes to and how it is degraded, the degree to which glycoGag is expressed is very important. Previous papers (PMID: 27879338 and PMID: 31964738) have shown that only a very small amount of glycoGag is required to counteract SERINC5/IFITM3 and rescue MLV infectivity. Conversely, transfection of elevated levels of glycoGag plasmid causes loss of virus infectivity for unknown reasons. This could be discussed in the final paragraph of the Discussion, when authors speculate about why glycoGag is rapidly turned over in proteasomes.

3. I disagree with the author’s interpretation of Figure 4I and 4J. The results show that CCNK knockdown and CDK13 shRNA are important for the downmodulation of Ser5 by glycoMA. The effect is partially but lanes 6 and 8 should be compared. There is a definite rescue of Ser5 protein by CCNK shRNA and CDK13 shRNA. Furthermore, the authors have duplicated a membrane in Figure 4I and 4J. On the right side, the Ser5 blots are identical in 4I and 4J. Even though the brightness/contrast of the two images is different, it is clearly the same membrane. This should be corrected through independent experiments.

4. The authors base a number of their conclusions on experiments utilizing 3-MA and Ly, which are non-specific inhibitors of PI3K family members as well as mTOR. The authors should include an experiment with SAR405, a specific inhibitor of PI3KC3 (vps34) and which inhibits production of PI3P, to confirm that macroautophagy is not responsible for turning over SERINC5 in glycoGag expressing cells. A paper that has successfully used SAR405 to dismiss a role for macroautophagy is PMID: 36264642.

5. The full dataset of the AP-MS needs to be provided in the supplemental information. The reviewers and readers should be able to see the full set of proteins shown to interact with Nef and glycoGag, as well as their relative abundances.

6. In Figure 3E, why is SERINC5 K130R cotransfected with RETREG1 (left part of figure) while SERINC5 WT is cotransfected with RETREG1-2 isoform? I think that SERINC5 WT should feature throughout Figure 3E.

7. I do not think that the authors do a convincing job of distinguishing between macro-ER-phagy and micro-ER-phagy. The authors claim in the Results and Discussion sections that their results with glycoGag suggest that it is using the micro-ER-phagy pathway to degrade SERINC5 WT and K130R, but they do not point to the specific data result that led them to that conclusion. I would suppose that they come to this conclusion because ATG5 and ATG7 are not required for glycoGag-mediated degradation of SERINC5. Since ATG5 and ATG7 are important for autophagosome function during macroautophagy, but it is not clear to me whether ATG5 and ATG7 are required for macro-ER-phagy. I would argue that it is not important for them to make the distinction between macro-ER-phagy and micro-ER-phagy when interpreting their results. It is sufficient to indicate that ER-phagy mediated by RETREG1 is the mechanism by which glycoGag antagonizes SERINC5.

8. It is possible that, if the authors studied various isolates of Nef and glycoGag, their findings could be different. To what extent is the use of ER-phagy by glycoGag conserved, for example? Perhaps some isolates of Nef also use ER-phagy to antagonize Ser5?

Reviewer #2: Fig. 1. Experiments should be reproduced in cells expressing full-length glycoGag.

Fig. 3B. No controls to show purity of the membrane fractions are included. Additionally, making conclusions on how SER5 affects the subcellular distribution of glycoMA by looking at subcellular fractionation assays is an overstatement. The authors should assess this by fluorescence microscopy and accompany the images with quantifications.

Fig. 4 F/H/I. The conclusions derived from these panels (that glycoMA downregulates SER5 from both the cell surface and cytoplasm) are an overstatement. The blots presented represent to whole cell lysates (not fractionations). Moreover, the results in panel G, where most SER5 appears in intracellular compartments in the presence of glycoMA, may represent an intermediate phenotype of glycoMA-dependent downregulation of SER5. To drive conclusions on whether glycoMA can downregulate SER5 present at intracellular membranes, authors should run experiments in the presence of inhibitors of ER to membrane transport or a similar approach.

Fig. 6B. Claims about RETREG1’s role in the downregulation of SER5 by glycoGag should be accompanied by quantifications from at least 3 independent experiments.

Statistics: no statistics are provided. Most of the data is derived from duplicate experiments.

Reviewer #3: In line 665, Figure 1A legend, MLV Gag is not translated from an alternatively spliced RNA but from a full length genomic RNA as is the case for all retroviruses. Please fix this.

In line 122, the authors state that the half-life of glycoGag, based on Figure 2B data, is 15 minutes. This is incorrectly stated. The authors in Figure 2B are using 0 and 2 hour time points and they lose signal at some point between these two time points. It is not clear how they draw the conclusion of 15 minutes from the data presented in 2B. This statement needs to be rewritten to accurately reflect the data shown.

For Figure 2C, the experiment is not described in the materials and methods, which made it difficult to follow at times. Furthermore, if they see no effect with 3MA, Ly and Baf, how do they know their treatment worked? Controls for this experiment are missing.

Figure 2D, the Cul3-KLHL20 E3 ligase is needed for glycoGag/glycoMA degradation in proteasomes in 293T cells. These viral proteins are found in MLV and as such are present in mouse cells in infections under physiological conditions. So, is this E3 ligase needed the same for proteasomal degradation of glycoGag/glycoMA under physiological conditions? The authors need to test this in mouse cells as they are more physiologically relevant than human cells and upon infection.

In Figure 3A, it would be better if localization of glycoGag/glycoMA was performed in murine cells. Also, the authors state that Nef is found in the plasma membrane but glycoGag/glycoMA are in the cytosol (lines 137-138). The authors should use markers to verify that Nef is on the plasma membrane in their system and it would be more informative if they defined where in the cytoplasm is glycoGag/glycoMA.

In Figure 3B, it is not clear why SER5 and glycoMA are in the cytosol fractions. The kit (Beyotime P0033-1) they are using is purifying total membranes in the cell (mitochondrial, Golgi, PM and ER) vs. cytosolic fraction which contains all proteins found in the cytosol (non-membrane bound proteins). How is glycoMA and SER5 in the cytosolic fraction when they have transmembrane domains? glycoMA and Ser5 should only be found in the membrane fraction.

In Figure 3B, which band is Nef and which one is glycoMA in the middle blot? The only bands shown are running at the same height, which is confusing as Nef and glycoMA are not of the same size (as shown in Fig. 3C. This blot is confusing.

In Fig3C, it would be more physiologically relevant if this coIP was performed with mouse SERINC5 and not the human ortholog.

In Figure 4B, the authors must explore the localization of mouse and human SER5 using markers for the different organelles with both wildtype and K130 or 131R. The authors stating in line 160-161 that wildtype SER5 is found on the plasma membrane is incorrect. This conclusion can only be drawn if they perform co-staining with a PM marker.

The majority of experiments in figure 4 that demonstrate differences between Nef and glycoMA on SER5 are performed in the context of transfections with either 293T cells or HeLas. Nothing is done in murine cells or in the context of infection. Moreover, panels 4F-J are performed using human SERINC5. Experiments using glycoMA should be performed with murine SER5 and not the human orthologs.

In figure 4F, control blots showing absence of KLHL20 and Cul3 are not provided and should be included in this figure panel.

Figure 4I and J, the requirement for CCNK and CDK3 with glycoMA is not absolute. While there is no complete rescue of SER5 in the case of glycoMA, when CCNK and CDK13 are knocked down, there is still more protein present in lanes 8 when compared to lanes 6. So the authors cannot state that CDK13 is not required for downregulation of SER5 (line 190). These findings suggest that there might be an additional factor involved along with CCNK and CDK13. This needs to be rewritten.

The authors state in lines 191-192 that the data “confirm that SER5 downregulation by GlycoGag does not necessarily require Ser5 on cell surface”. This is an overinterpretation of the data. The findings suggest that, but only if the authors block SER5 translocation to the PM, and still see reduction in SER5 levels, they can make that statement.

For Figure 5B, the authors are performing experiments on SER5 localization in HeLa cells with either Nef or GlycoMA. This is not the appropriate cell type or SER5 ortholog used. The authors ought to use a murine cell line and use mouse SER5 to say anything about the effect of GlycoMA on SER5 localization. Moreover, this experiment needs to be performed in the presence of MLV infection.

For Figure 5B and C, the authors need to employ additional quantitative approaches on their imaging to confirm LAMP1 or CALR colocalization.

Figure 5D legend is missing. In fact it says “Write down method:…”

There is a figure legend 5E, but there is no 5E figure.

For Figure 6, it would be beneficial for the authors to quantify band intensity. Also there is no mention in the figure legend about number of replicates for this experiment.

In Figure 6, there are panels that are missing their labeling. It is assumed that it is the ER-phagy receptors, but it would be easier to the reader to provide that next to the blot.

All data on Figure 7 are in HEK293T cells, which are not physiologically relevant to glycoGag and in extension to MLV. The authors need to do these experiments in murine cells using mouse SER5. Also, RETREG1 used in this assays is the human ortholog. It is never shown what happens with the murine ortholog and most importantly during infection in murine cells.

Figure 7A and C, there should be quantification of the SER5 bands in the different conditions, as that would be more convincing.

Looking at the SER5 western blot panels in Figure 7C, it shows that K130R SER5 is unaffected by the presence of glycoGag in lane 2. This is confusing, as all the data from prior figures (e.g., fig 7A and B) clearly show a reduction of K130R SER5 in the presence of glycoGag. The authors need to address that, as it contradicts their previous findings.

Figure 8, quantification of the SER5 bands showing the reduction of SER5 under the different conditions/knockout cells will add rigor to the data shown. Furthermore, Figure 8 would be better done in the context of MLV infection in murine cells, as it would show the effect of glycoGag on SER5 during microER-phagy under physiological conditions.

**Part III – Minor Issues: Editorial and Data Presentation Modifications**

Reviewer #1: Minor:

1. If authors intend to state that glycoGag and glycoMA exhibit an intracellular distribution, as opposed to localization to the plasma membrane, they should not use the term “cytoplasmic localization” to describe the location of glycoGag and glycoMA. As they show, these proteins colocalize with the ER marker CALR. As transmembrane proteins, they are membrane bound and therefore not technically “cytoplasmic.” It is better to remove that descriptor and indicate that they are found intracellularly in ER membranes.

2. “Transiently knocked down using CRISPR/Cas9” should be replaced with “knocked out using CRISPR/Cas9 in a population of cells.”

3. In Figure 3, HEK293T are mislabeled as “HEK393T.”

4. Can the authors comment on why they think SERINC5 protein levels increase in p62 KO and PI3KC3 KO cells? Also, why does SERINC5 protein level decrease upon ATG3 KO or ATG5 KO? These findings are unexpected and interesting.

5. Typo in Figure 6 (“glucoGag”)

6. Typo in Figure 8 (“knoctouts”)

Reviewer #2: Line 90 change “downregulates” by “downregulate”

Fig. 2. To confirm specificity, the authors should silence another E3 ligase complex

Fig. 3. Besides localizing at the plasma membrane, Nef is also present in the cytoplasm (also confirmed in their Fig. 2B data). The selection of images in Fig. 3A does not allow to see Nef’s cytoplasmic localization, probably because the cells selected are close to each other and the cytosol is much smaller than the panels for glycol-Gag and glyco-MA. Authors should select for cells with similar characteristics.

Fig. 3B. GlycoMA is detected also in the membrane fraction in cells expressing SER5. This needs to be acknowledged in the results section (lines 141-142). Also, add the ladder marker for the Nef and glycoMA membrane, as it is difficult to discriminate whether the bands correspond to Nef or glycoMA.

Fig. 4E. The authors should clarify that here both Nef and glycoMA have the same tag. Also, molecular weight indicators are encouraged throughout the manuscript.

Lines 145-146. None of the conditions in Fig. 3B have Nef and glycoMA expressed together. So, I am unsure what the authors are trying to claim when they say with the statement “when co-expressed with Nef and glycoMA, Ser5 did not alter the levels of Nef in the cytosolic and membrane fractions”.

Lines 263-264. The conclusion that glycoGag downregulates SER5 via LC3-dependent transport or CMA is an overstatement here. The authors should change the language saying that the findings in Fig 8A suggest that glycoGag downregulates SER5 via micro-ER-phagy. To make such conclusion, the authors should use drugs specific for micro-ER-phagy (3-MA, Baf-A1 nor Ly target this type of autophagy in a specific manner), or silence genes that are specific for this pathway other than RETREG-1.

Fig 5B. Staining for Nef and glycoMA is missing. Without detecting these proteins in the selected images, we cannot know if the cells selected were indeed Nef+ or glycoMA+.

Fig. 6. Each panel is lacking the labeling in one of the membranes (jusy above the actin membrane). I am assuming this corresponds to each ER-phagy receptor being tested, but this needs to be labeled.

Fig. 7E. Without adding a beads-only and an IgG-only control, the authors cannot conclude that RETREG1 is only present in the pulldown fraction of cells co-expressing SER5 and glycoGag, as there are bands with similar molecular weight in the SER5 only and SER5 + Nef samples.

Fig. 8. The authors need to provide evidence showing lack of expression of target genes in the knockout cells

Reviewer #3: (No Response)

PLOS authors have the option to publish the peer review history of their article (what does this mean? ). If published, this will include your full peer review and any attached files.

**Do you want your identity to be public for this peer review?** For information about this choice, including consent withdrawal, please see our Privacy Policy .

Reviewer #1: No

Reviewer #2: No

Reviewer #3: No

---

## [Decision Letter · Decision Letter 1]

15 Jul 2025

Murine Leukemia Virus GlycoGag Antagonizes SERINC5 via ER-phagy Receptor RETREG1

PLOS Pathogens

Dear Dr. Zheng,

Thank you for submitting your manuscript to PLOS Pathogens. After careful consideration, we feel that it has merit but does not fully meet PLOS Pathogens's publication criteria as it currently stands. Therefore, we invite you to submit a revised version of the manuscript that addresses the points raised during the review process.

Please submit your revised manuscript within 60 days Sep 13 2025 11:59PM. If you will need more time than this to complete your revisions, please reply to this message or contact the journal office at plospathogens@plos.org. Please include the following items when submitting your revised manuscript:

We look forward to receiving your revised manuscript.

Kind regards,

Welkin E. Johnson

Academic Editor

PLOS Pathogens

Richard Koup

Section Editor

PLOS Pathogens

Editor-in-Chief

PLOS Pathogens

orcid.org/0000-0003-2946-9497

Michael Malim

Editor-in-Chief

PLOS Pathogens

orcid.org/0000-0002-7699-2064

**Journal Requirements:**

1)  Please ensure that the funders and grant numbers match between the Financial Disclosure field and the Funding Information tab in your submission form. Note that the funders must be provided in the same order in both places as well.  

**Reviewers' Comments:**

Reviewer's Responses to Questions

**Part I - Summary**

Reviewer #4: The authors reported a new mechanism (via ER-phagy) for GlycoGag to down regulate serinc5. The experiments are well designed and the results support authors' major conclusion. The report should excite people who study host-virus interactions.

This reviewer does have some concerns.

Reviewer #5: Overall, the paper is rich with data and the findings are interesting. They appear to have significance for the mechanism of Ser5 downregulation by the MLV glycoGag protein. The impact may be limited by the observation that RETREG1 appears at least partially dispensable for downregulation of wild type Ser5. Perhaps it would be informative to perform assays of glycoMA rescue of HIV-1 infectivity from Ser5 inhibition in a series of cotransfection assays in RETREG1-deficient vs. control cells.

**Part II – Major Issues: Key Experiments Required for Acceptance**

Reviewer #4: 1. It seems glycoGag directs non-cell-surface (ER) serinc5 for degradation via ER-phagy and directs cell-surface serinc5 “likely …. in endolysosomes”. Because it is the cell-surface serinc5 that inhibits virus replication, the authors should examine the real contributions of ER-phagy in serinc5 antagonism by glycoGag. The authors can perform the virus infectivity assay (as in Figure 1B) in RETREG1 KO cells to test to which extent the KO could reduce glycoGag’s antagonism activity.

2. All experiments were performed using over-expressed glycoGag. I agree with reviewer#3 that it is important to test glycoGag’s function at physiological level. The authors can infect WT and RETREG1 KO NIH3T3 cells with MLV (as in Figure 2A) and then examine the degradation of WT and K130R serinc5.

Reviewer #5: 1. Fig. 1A. Authors have concluded that glyco-Gag adopts a (unexpected) topology in the membrane based on amino acid sequence alone (line 103). This could be correct, but in the absence of confirmation (e.g. by staining of cells with an MLV CA antibody), seems speculative.

2. Fig. 4I. Authors claim that glycoMA can downregulate Ser5 under conditions in which Nef cannot. In these experiments, the ratios of levels of glycoMA vs. Nef seem much greater than those in other experiments (e.g. 4E). In these experiments, were cells transfected with the same amounts of glycoMA and Nef expression plasmids? If so, then why do the levels appear to be so different?

3. Fig. 4J. Authors draw conclusions regarding downregulation of Ser5 mutants from images of one or two cells. This falls short of establishing reproducibility. Also, Ser5.S360A levels appear to vastly exceed those of WT and S249A in Fig. 4H. Is the apparently lack of downregulation simply owing to excessively high levels of the S360A mutant?

4. Fig. 5D: It would be helpful to confirm the key associations detected by MS by IP-western.

**Part III – Minor Issues: Editorial and Data Presentation Modifications**

Reviewer #4: 1. The expression of serinc5 K130R in RETREG1 KO cells is not consistent in Figure 7A and Figure 7D. In 7A, serinc5 is expressed at obviously lower level in KO cells, but in 7D, the expression levels in WT and KO cells are comparable. Is this due to different transfection efficiency in different experiments?

2. Figure 3C. The authors concluded that “glycoGag is targeted to proteasomes for degradation”. However, in Figure 5A, proteasome inhibitor MG132 treatment did not increase the levels of glycoGag.

3. Figure 5 showed that glycoGag strongly localized in ER. Then why glycoGag cannot be detected in the membrane fraction (Figure 3B).

4. The authors stated that “serinc5 significantly increased the levels of glycoMA in the membrane fraction” (line 145). I am wondering why only serinc5, but not glycoGag, is degraded in the membrane fraction.

5. Line 240, “K130” should be “K130R”.

Reviewer #5: 1. The abstract could be improved by rewriting for a broader audience. I would remove the phrases “employing the K130R mutant” (line 27); “while still in the cytoplasm” (line 30). I also found the sentence in lines 30-32 to be confusion: are the authors claiming that SERINC5 binds glycoGag in the cytoplasm and moves it to the plasma membrane, only to be downregulated by it? This seems paradoxical. I would also de-emphasize the method of MS in the abstract, as this is an unnecessary detail. Finally, the authors end both the Abstract and Discussion with a tag line about the arms race between retroviruses and Ser5. What exactly are these “critical new insights?” Aren't they just mechanistic details of Ser5 downregulation by different viral proteins?

2. Fig. 1C. Authors directly compare levels of Nef and glycoMA based on band intensities on an immunoblot. Presumably, both bands were detected by an HA-specific antibody, but this was not stated.

3. Line 200: change to: “…require Ser5 accumulation at the cell surface.”

4. Line 214: change to: “…but also showed some intracellular colocalization with CALR.”

5. Lines 285-286: Authors should comment on the apparent stabilization of glycoMA by 3-MA. What does this suggest regarding PI3KC3 and/or autophagy and glycoMA stability?

PLOS authors have the option to publish the peer review history of their article (what does this mean? ). If published, this will include your full peer review and any attached files.

**Do you want your identity to be public for this peer review?** For information about this choice, including consent withdrawal, please see our Privacy Policy .

Reviewer #4: No

Reviewer #5: No

**Figure resubmission:**

**Reproducibility:**



---

## [Editor Report · Decision Letter 2]

15 Aug 2025

Dear Dr. Zheng,

We are pleased to inform you that your manuscript 'Murine Leukemia Virus GlycoGag Antagonizes SERINC5 via ER-phagy Receptor RETREG1' has been provisionally accepted for publication in PLOS Pathogens.

Best regards,

Welkin E. Johnson

Academic Editor

PLOS Pathogens

Richard Koup

Section Editor

PLOS Pathogens

Sumita Bhaduri-McIntosh

Editor-in-Chief

PLOS Pathogens

orcid.org/0000-0003-2946-9497

Michael Malim

Editor-in-Chief

PLOS Pathogens

orcid.org/0000-0002-7699-2064
---

## [Editor Report · Acceptance letter]

Dear Dr. Zheng,

We are delighted to inform you that your manuscript, " 

Murine Leukemia Virus GlycoGag Antagonizes SERINC5 via ER-phagy Receptor RETREG1," has been formally accepted for publication in PLOS Pathogens.

Best regards,

Sumita Bhaduri-McIntosh

Editor-in-Chief

PLOS Pathogens

orcid.org/0000-0003-2946-9497

Michael Malim

Editor-in-Chief

PLOS Pathogens

orcid.org/0000-0002-7699-2064